# Single-shot observation of optical rogue waves in integrable turbulence using time microscopy

Pierre Suret[1,2], Rebecca El Koussaifi[1,2], Alexey Tikan[1,2], Clément Evain[1,2], Stéphane Randoux[1,2], Christophe Szwaj[1,2] & Serge Bielawski[1,2]

Optical fibres are favourable tabletop laboratories to investigate both coherent and incoherent nonlinear waves. In particular, exact solutions of the one-dimensional nonlinear Schrödinger equation such as fundamental solitons or solitons on finite background can be generated by launching periodic, specifically designed coherent waves in optical fibres. It is an open fundamental question to know whether these coherent structures can emerge from the nonlinear propagation of random waves. However the typical sub-picosecond timescale prevented—up to now—time-resolved observations of the awaited dynamics. Here, we report temporal 'snapshots' of random light using a specially designed 'time-microscope'. Ultrafast structures having peak powers much larger than the average optical power are generated from the propagation of partially coherent waves in optical fibre and are recorded with 250 femtoseconds resolution. Our experiment demonstrates the central role played by 'breather-like' structures such as the Peregrine soliton in the emergence of heavy-tailed statistics in integrable turbulence.

[1] Laboratoire de Physique des Lasers, Atomes et Molecules, UMR-CNRS 8523, Université de Lille, Villeneuve d'Ascq 59655, France. [2] Centre d'Etudes et de Recherches Lasers et Applications (CERLA), Villeneuve d'Ascq 59655, France. Correspondence and requests for materials should be addressed to P.S. (email: Pierre.Suret@univ-lille1.fr).

First studied in oceanography, rogue waves (RWs) also called freak waves are waves of giant amplitude that occur more frequently than expected from the normal law[1–3]. From the pioneering optical fibre experiment performed by Solli et al in 2007 (ref. 4), optical RWs have been studied in various optical experiments such as supercontinuum generation in fibres[4–7], laser filamentation[8,9], passive cavities[10], lasers[11–14] and Raman fibre amplifiers[15]. In all these experiments, the so-called optical RWs (and more generally extreme events) correspond to a large variety of phenomena, often not comparable to ocean waves, and their generation cannot be ascribed to a unique physical mechanism[1,3,16–18].

In this context, however, the one-dimensional focusing nonlinear Schrödinger equation (1D-NLSE) plays a fundamental role by providing a one-to-one correspondence between some uni-directional systems in hydrodynamics and in nonlinear optics[19–21]. The focusing 1D-NLSE indeed describes at leading order propagation of deep water waves in one-dimensional tanks and also nonlinear propagation of light in single-mode fibres[21]. Moreover it admits breathers solutions that exhibit localization and amplitude properties having some degree of compatibility with the definition of RW objects[22–26]. It is conjectured by some authors that breather solutions of the 1D-NLSE such as the Peregrine soliton (PS) may represent prototypes of RWs[20,22,24,26–28]. This has motivated nice experiments in which these solitons on finite background (SFB) have been generated in optical fibres[29–32] and in a one-dimensional water tank[32,33]. In these experiments, SFB are generated from coherent and deterministic (that is, non-stochastic) initial conditions that have to be carefully designed.

Alternatively, hydrodynamical and optical fields can be of a random and irregular nature. The questions related to the local emergence of coherent structures compatible with breathers solutions of the 1D-NLSE are still completely open in the context where waves randomly fluctuate in space or time. The problem of random (or partially coherent) inputs in wave systems described by integrable equations such as 1D-NLSE enters within the fundamental framework of the so-called 'integrable turbulence' introduced by V.E. Zakharov[34–39]. As wave interactions are not resonant for the 1D-NLSE[40,41], features characterizing integrable turbulence are of profoundly different nature than those found in conventional turbulence and they are still not well understood[36,39,40,42].

Partially coherent waves correspond here to random waves whose optical spectral width is finite and small in comparison with the carrier wave frequency[43]. Assuming independent random Fourier components the statistics of the partially coherent field is Gaussian[43]. It is now well established that heavy-tailed deviations from Gaussianity can occur in integrable turbulence. This phenomenon arises in particular when partially coherent waves are used as initial conditions in a wave system that is described at leading order by the focusing 1D-NLSE. This has been recently shown in optical fibre experiments reported in refs 38,42. Heavy-tailed deviations from Gaussianity reported in previous experiments possibly arise from the stochastic generation of SFB that are localized in space and time, such as the PS. So far, this conjecture is supported by numerical simulations of the 1D-NLSE that have shown that some high-amplitude coherent structures compatible with some breather solutions of the 1D-NLSE like the PS can spontaneously emerge from a stochastic background[25,28,36,38].

However, the stochastic and local generation of SFB solutions of the 1D-NLSE has never been reported so far. Moreover, the relationship between these structures and heavy-tailed statistics has never been experimentally demonstrated. Optical fibres represent suitable tabletop laboratory to generate deterministic

solutions of 1D-NLSE on the one hand[29,31,32,44] and investigate integrable turbulence on the other hand[38,42,45]. However, in optical fibres, typical time scale characterizing SFB or soliton solutions of 1D-NLSE falls in the picosecond range. Some well-established methods like intensity autocorrelation or frequency-resolved optical gating have been used to measure phase and/or power profiles of breathers having this duration[29,31]. However all those standard methods rely on the fact that the optical signal under investigation must be periodic. These methods are inherently inadequate for the precise observation of the dynamics of breather-like coherent structures that may emerge from the nonlinear propagation of non-periodic random waves.

From the technical point of view, it is important to note that, up to now and despite the numerous experiments devoted to optical RWs, the precise time domain observation of coherent structures embedded in random fields and compatible with 1D-NLSE solutions was not possible. Contrary to the direct observation of spatial structures in experiments in which the intensity profile of a light beam is recorded with a camera[9,46,47], the fast time scales of fluctuations (picoseconds or less) involved in single-mode fibre experiments make single-shot recording of RWs a particularly challenging task. Pioneer works performed in optical fibres hence did not provide single shot observation of RW objects but evidence of it by using, for example, spectral filtering[4,6,48] or statistical measurement from optical sampling techniques[38]. In the case of mode-locked laser dynamics, statistical measurement of the pulse amplitude can also been performed with standard photodetectors[12]. However, up to now, in all these experiments, the real-time and single-shot observation of the shape of the picosecond time scale structures generated from the nonlinear propagation of random waves has never been reported.

In this article, we present direct single-shot recordings of optical RWs by using a specially designed time microscope (TM) ultrafast acquisition system[49–51]. The temporal resolution of ∼250 fs of our TM (see Methods and Supplementary Note 1) allows us to investigate the fast dynamics underlying integrable turbulence arising in optical fibres. We study experimentally the changes of the dynamics and of the statistics arising from the deterministic and nonlinear propagation of partially coherent waves in optical fibre. The observations performed with our TM reveal the frequent emergence of subpicosecond structures having peak powers much larger than the average optical power. The field used as initial condition has a Gaussian statistics and coherent subpicosecond structures growing from nonlinear propagation induce heavy-tailed deviations from Gaussian statistics.

## Results

**Experimental setup and observation of rogue waves**. The principle of our experiments is displayed in Fig. 1a. Partially coherent waves are launched into an optical fibre. Observations performed with the TM (see Fig. 2) at the output of the fibre immediately reveals the emergence of intense peaks, with powers frequently exceeding the average power $\langle P \rangle$ by factors of 10–50 (see Fig. 1b–d and Supplementary Movie 1). Starting from random fluctuations having typical time scale around 5–10 ps (Fig. 1a), those extreme events are also found to be extremely short, with time scales of the order of several hundreds of femtoseconds (Figs 1b–d and 3b–d).

More precisely, the random waves used as initial conditions in our experiments are partially coherent light waves emitted by a high power Amplified Spontaneous Emission (ASE) light source at a wavelength $\lambda \sim 1560$ nm (see Fig. 1). Using a programmable

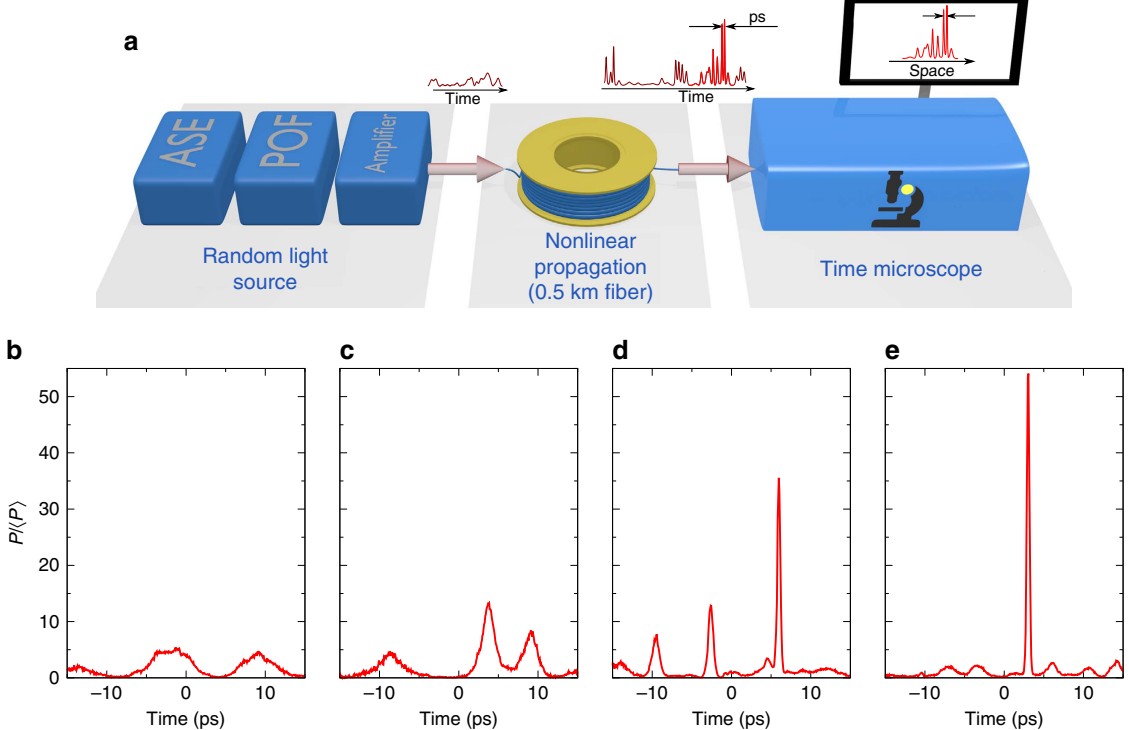

**Figure 1 | Overview of experiments.** (**a**) Global strategy for the experimental observation of the fast dynamics. Incoherent light from a 1560 nm Amplified Spontaneous Emission (ASE) source is filtered by using a Programmable Optical Filter (POF) and amplified before experiencing nonlinear propagation in a polarization maintaining fibre. Single-shot measurement of random light is achieved by using a specially designed time-microscope (TM). The TM, which maps the temporal evolution onto the spatial coordinate of a sCMOS camera, has a temporal resolution of 250 femtoseconds (see Fig. 2 and Methods for details). (**b**–**e**) Typical single-shot recordings of the fast dynamics of optical power (normalized by its mean value). Initial spectral width $\Delta v = 0.1$ THz. (**b**) Initial condition. (**c**–**e**) Structures observed at the output of the fibre for mean powers $\langle P \rangle = 0.5$ W (**c**), $\langle P \rangle = 2.6$ W (**d**), $\langle P \rangle = 4.$ W (**e**).

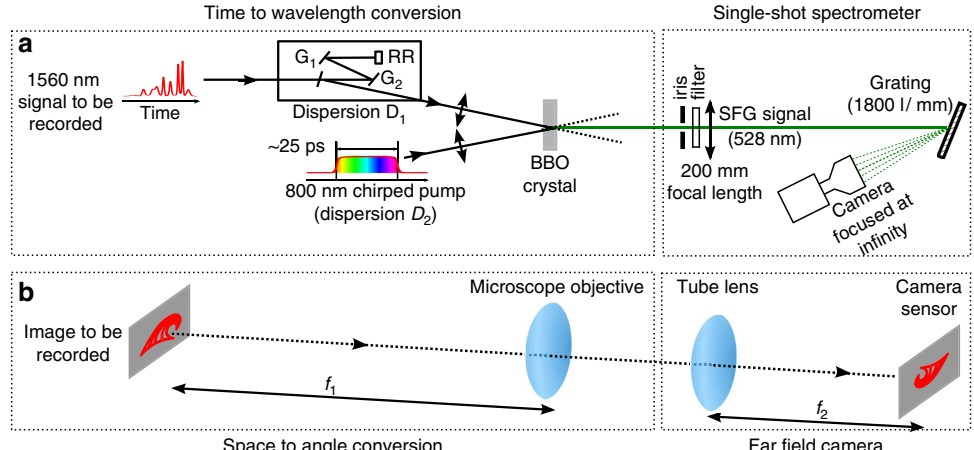

**Figure 2 | Time microscope realized for the ultrafast observation of optical rogue waves.** (**a**) Experimental setup. A key element is the time lens, which is composed of the BBO crystal, pumped by the stretched 800 nm pulse. (**b**) Spatial analog of the time microscope. The dispersion $D_1$ (provided by the grating compressor) is analogue to the diffraction between the initial image and the lens with focal length $f_1$. The time lens is the analog of the ($f_1$) lens. The single-shot spectrometer is formally analogue to the far-field camera. $G_1$ and $G_2$ are 600 lines mm$^{-1}$ gratings, RR is a roof retroreflector. Note that the BBO crystal is placed at the focal plane of the 200 mm collimating lens. Transport optics are not represented.

optical filter, the optical spectrum of the partially coherent light is precisely designed to assume a Gaussian shape having a full width at half maximum that is adjusted either to $\Delta v = 0.1$ THz or $\Delta v = 0.05$ THz. The partially coherent waves are launched into a 500 m long single mode polarization maintaining fibre at a wavelength falling into the anomalous (focusing) regime of dispersion. The light at the output of the nonlinear fibre is then

directed to the TM (detailed in Fig. 2), which acquires traces (optical power versus time over a $\approx 20$ ps long window) at a rate of 500 per second, and displays the signals in real time.

As in a standard spatial imaging microscope, the TM is composed of an objective and a tube lens (see Fig. 2b). The objective is a time lens[49–51] operating from sum-frequency generation (SFG) between the 1560 nm signal and a chirped

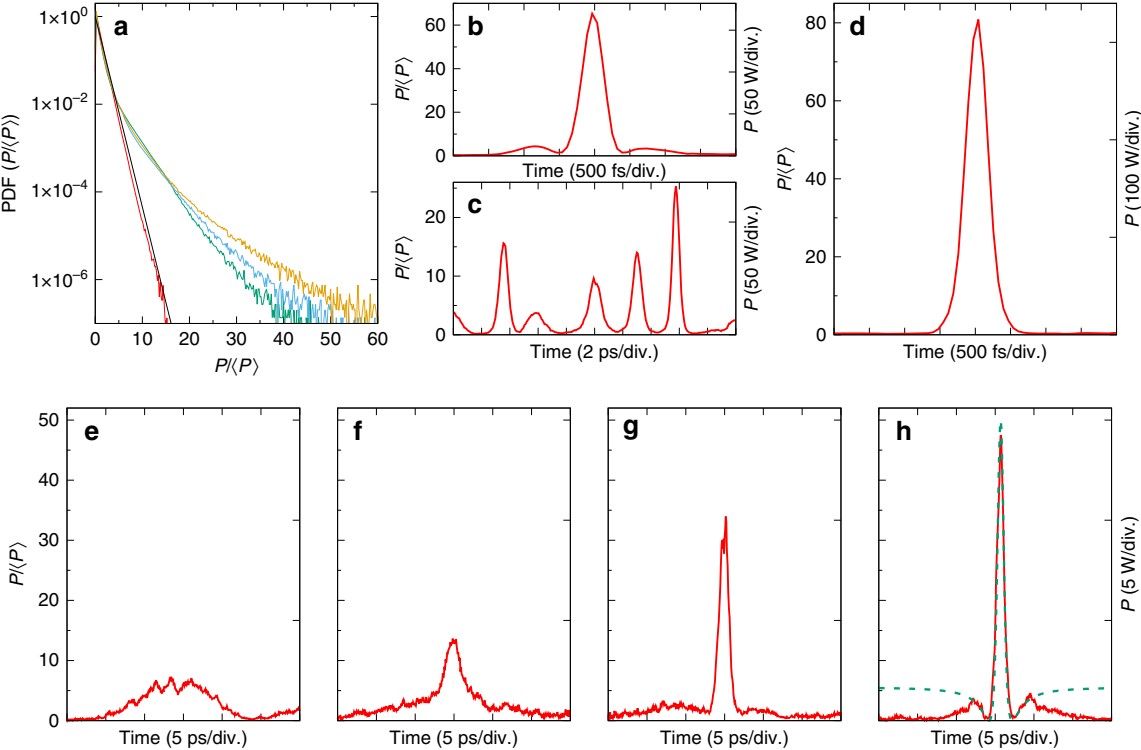

**Figure 3 | Typical temporal dynamics and statistics (experiments). (a)** Probability density function (PDF) of the initial condition (red line) and of the output power at 500 mW (green line), 2.6 W (blue line), 4 W (yellow line) with an initial spectral width $\Delta v = 0.1$ THz. The black line represents the normalized exponential distribution : $\text{PDF}(P/\langle P \rangle) = \exp(-P/\langle P \rangle)$. **(b–d)** Typical random signals recorded by the TM at the output of the nonlinear PM fibre. **(b)** PS-like structure ($\langle P \rangle = 2.6$ W, $\Delta v = 0.1$ THz). **(c)** Series of multiple peaks ($\langle P \rangle = 2.6$ W, $\Delta v = 0.1$ THz). **(d)** Giant optical RW reaching a peak power $\sim 80$ times greater than the mean power of the random wave ($\langle P \rangle = 4$ W, $\Delta v = 0.1$ THz). **(e–h)** Typical snapshots showing several stages leading to the emergence of a breather-like structure ($\langle P \rangle = 300$ mW, $\Delta v = 0.05$ THz). The green dashed line in **(h)** represents the best fit to the analytical expression of a PS.

pump pulse (at 800 nm) (see Methods). The observation in the focal plane of the tube lens is achieved by a spectrum analyser (composed of a diffraction grating, a lens and a camera). The use of this TM strategy enables to easily reach extremely high dynamical ranges (up to 40 dB; see Methods), which is a crucial point for analysing extreme events embedded in moderate power fluctuations. The temporal resolution of our TM is $\sim 250$ fs (see Methods, Supplementary Note 1 and Supplementary Figs 1–3).

**Evidence of heavy-tailed statistical distributions.** In order to quantify the emergence of high power structures, we compute statistical distributions from a large amount of data recorded with the TM at the input and at the output ends of the fibre (see Methods). As expected, the probability density function (PDF) of the optical power emitted by the ASE source is systematically very close to the exponential distribution. The exponential distribution of the power $|\psi|^2$ (see Fig. 3a) corresponds to a Rayleigh distribution of $|\psi|$ and to a Gaussian statistics for the field $\psi$ under the assumption of uniformly distributed random phases[1,36,38,43] (See Supplementary Note 2 and Supplementary Fig. 4). According to the central limit theorem, the Gaussian statistics of the field used as initial condition arises from the linear superposition of numerous independent waves (that is, independent spectral components of the ASE source). On the contrary, the PDF of light power measured at the output of the nonlinear fibre is found to exhibit heavy-tailed deviations from the exponential distribution, thus confirming the generation of RWs (see Fig. 3a). Moreover, the PDFs demonstrate that the number of structures having high

peak power increases while the mean power $\langle P \rangle$ of random optical waves (that is, the strength of nonlinearity) increases (see Fig. 3a).

**Analysis of coherent structures.** Experimental signals plotted in Figs 1a–d and 3b–h represent direct and accurate observations of the dynamical structures underlying these heavy-tailed statistics. Starting from random light propagating with a mean power of 4 W in the fibre, huge RWs having peak power that exceeds 300 W can be observed at the output of the fibre (Fig. 3d). Note that the conventional threshold of RWs in the experiments performed with $\langle P \rangle = 4$ W is given by $P_{RW}/\langle P \rangle \simeq 9.7$ (see Supplementary Note 2 and Supplementary Fig. 4).

From a careful analysis of the data, two typical shapes can be distinguished : isolated breather-like structures (see Fig. 3b and Supplementary Movie 1) and more complicated structures composed of several peaks (see Fig. 3c and Supplementary Movie 1). It is difficult to draw clear conclusions from single-shot observation displayed in Fig. 3c: these structures might be compatible with interacting fundamental solitons or with randomly perturbed periodic Akhmediev breathers. On the contrary, some of the single-shot observations seem to correspond to different dynamical stages that are typical for the emergence of 'breather' solutions of 1D-NLSE. In order to illustrate the process of emergence of isolated breather-like structures, we plot in Fig. 3e–h different structures observed after the propagation of partially coherent waves having an initial spectral width $\Delta v = 0.05$ THz and an average power $\langle P \rangle = 0.3$ W.

The precise understanding of the mechanisms underlying these dynamical features (Fig. 3b–h) is an open question. Our

experiments are performed in a strongly nonlinear regime (see Methods) and they are compatible with the emergence of 'breather-like' dynamics associated to the N-soliton solutions of 1D-NLSE with zero boundary conditions[52]. Moreover, we have selected structures displayed in Fig. 3e–h because their shapes are strikingly similar to those found in the theoretical scenario demonstrated in the semi-classical (small-dispersion) regime : the evolution of a single hump leads in a generic way to the emergence of a Peregrine breather[53]. Partially coherent fields with narrow spectrum and having a Gaussian statistics at initial stage consist of many 'slow' fluctuations that are individually compatible with such humps (see Figs 1a and 4b). Remarkably, the *power* profile of the exact analytical PS coincides with some of the structures experimentally observed (see green dashed line in 3(h)). However note that the precise identification of the breather-like structures (as PS or Akhmediev breather or other more complex solutions of 1D-NLSE) would require a simultaneous measurement of the phase dynamics[37].

**Numerical simulations.** Behaviours observed in experiments can be well reproduced from numerical simulations of the 1D-NLSE (see Methods). First of all, the PDFs of optical power (see Fig. 4a) and the optical spectra (see Supplementary Fig. 5 and Supplementary Note 3) are well reproduced by numerical simulations. Figure 4 shows a picture of typical random fluctuations of the optical power that are found at the input and output ends of the optical fibre. Taking a partially coherent light field having a bandwidth of 0.1 THz at initial stage, the typical time scale for power fluctuations is around a few picoseconds (Fig. 4b). The scenari observed in the experiments are also found in numerical simulations (see Fig. 4 and Supplementary Movie 2). In particular, either breather-like structures appear and disappear along the propagation (see Fig. 4g), either several pulses simultaneously emerge together from the random background (see Fig. 4c). Our experiments provide snapshots randomly recorded while the numerical simulations allow to follow the dynamics of nonlinear random waves along the propagation. In this respect, the numerical simulations reveal that the breather-like structures often emerge on the top of the initial power fluctuations (see Fig. 4d–g and Supplementary Movie 2).

## Discussion

We believe that our experiments bring new and fundamental data allowing to examine model and conjectures related to nonlinear random waves and integrable turbulence. In particular, in the past few years, it has been conjectured that breather solutions of 1D-NLSE such as PS or Akhmediev breathers represent prototypes of RWs[20,24,27–30,33]. These SFB solutions have been generated in a deterministic way in optical fibres[29–31] and in a one-dimensional water tank[33,54]. In ref. 54, it is also shown that the deterministic generation of the PS is robust against some additional noise. However, the relevance of the possible emergence of soliton on finite background solutions from a randomly fluctuating background is still discussed.

Our experiments with randomly fluctuating optical waves show that Peregrine-like breathers tend to emerge at the top of power fluctuations. These results may be interpreted in the light of recent mathematical results which show that PS spontaneously emerges in a generic way from an isolated hump[53]. In the context of integrable turbulence, our experiments thus confirm the relevance of a scenario in which the RW objects that spontaneously emerge from a noisy background are indeed locally compatible with breather solutions of the 1D-NLSE. Moreover and for the first time, our time-resolved observations correlate in an unambiguous way the occurrence of a heavy-tailed statistics with the frequent occurrence of breather-like coherent structures. The simultaneous fast measurement of phase and amplitude fluctuations represents the next experimental bottleneck for the careful identification of optical RWs.

The conclusions which are drawn here in optical fibre experiments well described by the 1D-NLSE cannot be directly extrapolated to phenomena found in the oceans. Let us emphasize however, that the principle of our experiment is identical to the one of some previous experiments performed in one-dimensional

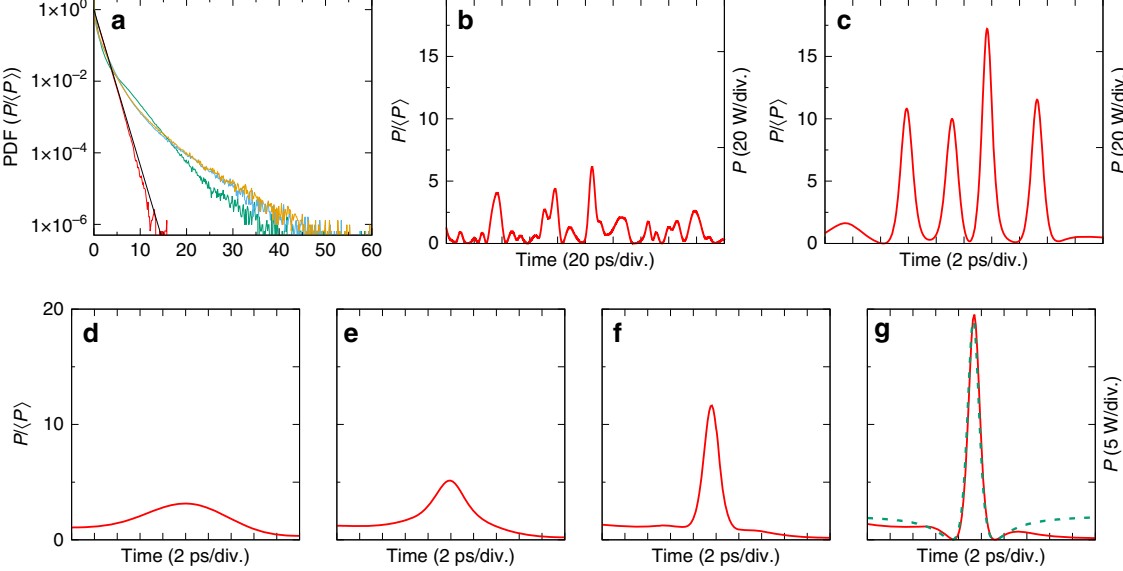

**Figure 4 | Typical temporal dynamics and statistics (numerical simulations of the 1D-NLSE).** (**a**) Probability Density Function (PDF) of the initial condition (red line) and of the output power at 500 mW (green line), 2.6 W (yellow line), 4 W (blue line). The black line represents the normalized exponential distribution : $PDF(P/\langle P \rangle) = \exp(-P/\langle P \rangle)$. (**b**) Typical fluctuations of the power of the random field used as initial condition. (**c**) Series of multiple peaks (zoom) ($\langle P \rangle = 2.6$ W, $\Delta \nu = 0.1$ THz). (**d-g**) Typical snapshots (zoom) showing several stages leading to the emergence of a breather-like structure along the propagation ($\langle P \rangle = 1$ W, $\Delta \nu = 0.05$ THz, $z = 0$ m (**d**), $z = 285$ m (**e**), $z = 415$ m, (**f**) $z = 500$ m). The green dashed line in (**g**) represents the analytical expression of a PS.

deep water tank[55]. Starting from random initial conditions, those hydrodynamical experiments have also demonstrated the formation of heavy-tailed statistics. Note finally that the extreme amplitudes observed in our optical fibres experiments could not be observed in water waves that are strongly limited by the wave breaking phenomenon[1,56,57].

The time-resolved direct observation of RWs presented in this letter opens the way to numerous other studies on fast dynamics in Optics. Integrable turbulence[36,38,39] is a fundamental framework to study nonlinear random waves because various systems such as one-dimensional deep-water waves or nonlinear optical waves are ruled at the leading order by the integrable 1D-NLSE[21,22,58]. One of the most intriguing issues of integrable turbulence is the dramatic dependence of the asymptotic statistics on the statistics of the initial condition[36,38,39]. On the one hand, and as it is shown in this article, initial conditions made of partially coherent waves lead to the emergence of strongly heavy-tailed PDFs[38]. On the other hand, the modulational instability is known to lead to a stationary state characterized by Gaussian statistics[36]. The difference between the nature of the structures emerging in the context of random waves on one hand or in the context of modulation instability on the other hand is an open and fundamental question[38,39]. The dynamics underlying the process of noise-driven modulational instability, of its nonlinear stage and more generally of optical turbulence represent open and fundamental questions that can now be experimentally studied by using our TM[36,59,60].

## Methods

**Partially coherent source and nonlinear propagation setup.** The partially coherent light (that is, the initial condition) is generated by an Erbium fibre broadband ASE source (from Highwave), which is spectrally filtered (with programmable shape and linewidth) using a programmable optical filter (Waveshaper 1000S, from Finisar). The output is then amplified by an Erbium-doped fibre amplifier (from Keopsys). This random light is launched into a single-mode polarization maintaining fibre (Fibercore HB-1550T), with 500 m length, a dispersion $\beta_2 = -20 \, ps^2 \, km^{-1}$ (measured). For a given spectral width, the power of the light launched inside the fibre is controlled using a half wavelength plate and a polarizing cube.

The experiments presented in the article are performed in nonlinear regimes characterized by a dispersion length $L_D$ much larger than the nonlinear length $L_{NL}$ in the initial stage. Typically, for a time scale $T_0 \sim 10 \, ps$ corresponding to the initial spectral width $\Delta\nu = 0.1 \, THz$, $L_D = T_0^2/|\beta_2| \sim 5 \, km$. Considering a hump with a maximum power $P = 5 \times \langle P \rangle$ and $\langle P \rangle = 0.5 \, W$ (Fig. 1b), the nonlinear length is $L_{NL} = 1/(\gamma P) \sim 0.2 \, km$.

**Time-microscope setup and performances.** For the single-shot acquisition of the subpicosecond optical signals, we realized an upconversion time-microscope, largely based on the work of ref. 50. From the input-output point of view, the TM encodes the temporal shape of the optical signal onto the spectrum of a chirped pulse (that is, spectral encoding). Then the spectrum is recorded using a simple spectrometer composed of a 1800 lines mm$^{-1}$ grating and a sCMOS camera. A region of interest (of typically 2048 × 8 pixels) is selected for recording the image (a raw image is presented in Supplementary Fig. 2).

For reaching high temporal resolution, a key element is the time lens[49], which is composed of a BBO crystal, pumped by a chiped 800 nm pulse. Before entering the time lens, the 1560 nm signal experiences anomalous dispersion in a classic Treacy grating compressor (see Fig. 2).

As in other time lens systems[49–51] high resolution requires proper adjustment of the 1560 nm compressor (see Supplementary Note 1 for adjustment detail, and performances of the setup). Conceptually, this is exactly analogue to the tuning of the object-microscope objective distance in classical microscopes. For all results presented in this paper, the temporal resolution is 250 fs FWHM (see Supplementary Figs 1 and 2), and the field of view is of the order of 20 ps. Temporal calibration of the time-microscope is obtained by recording the response to a series of two laser pulses with known delay (see Supplementary Fig. 3 and Supplementary Note 1).

As another crucial point, the time-microscope strategy leads to an extremely high dynamical range (that is, the ratio between maximal recordable signal and dark noise). This directly stems from the choice of employing a camera for the recording. More precisely, our 16-bit sCMOS camera has an RMS dark noise

of ≈2 electrons and a saturation value of 30,000 electrons, leading to a ≈40 dB dynamical range.

The 800 nm pump is provided by an amplified Titanium-Sapphire laser (Spectra Physics Spitfire, 2 mJ, 40 fs, a spectral bandwidth of about 25 nm), operated at 500 Hz, and only 100 nJ are typically used here. For inducing (normal) dispersion on the 800 nm pulses we simply adjusted the amplifier's output compressor. The dispersion was fixed to 0.23 ps$^2$, leading to chirped pulses of duration of about 20 ps. The 1560 nm grating compressor uses two 600 lines mm$^{-1}$ gratings, operated at an angle of incidence of 40 degrees, and whose planes are separated by 42 mm. The BBO crystal has 8 mm length and is cut for non-colinear type-I SFG (θ = 24.2°, φ = 90°, external angle between pump and signal = 12.5°). Focusing of the 800 and 1560 nm signals on the BBO crystal are performed by two lenses with 20 cm focal lengths. In order to improve the rejection of the 800 nm and the 1560 nm and to keep only the SFG at 528 nm, a 40-nm bandpass filter around 531 nm (FF01 531/40-25 from Semrock) is added after the crystal. The camera is a sCMOS Hamamatsu Orca flash 4.0 V2 (C11440-22CU), equipped with an 60 mm lens (Nikkor Micro 60 mm f/2.8 AF-D). The objective is focused at infinity and the waist of the SFG in the BBO crystal is imaged on the camera sensor. The camera is synchronized on the 800 nm laser pulses, and the integration time is adjusted to 1 ms, thus enabling single-shot operation of the time-microscope. PDFs of optical power are computed with 75.10$^6$ samples (10$^2$ points taken in the centre of the temporal field of the TM from 75.10$^4$ frames) for a given set of parameters.

**Numerical simulations details.** Numerical simulations are performed by integrating the 1D-NLSE :

$$i\frac{\partial\psi}{\partial z} = \frac{\beta_2}{2}\frac{\partial^2\psi}{\partial t^2} - \gamma|\psi|^2\psi \quad (1)$$

where $\psi$ is complex envelope of the electric field, normalized so that $|\psi|^2$ is the optical power, $z$ is the longitudinal coordinate in the fibre, and $t$ is the retarded time. $\beta_2 = -20 \, ps^2 \, km^{-1}$ is the second-order dispersion coefficient of the fibre and $\gamma$ is the Kerr coupling coefficient. From the comparison between the optical spectra measured in the experiments and the ones computed from the numerical simulations, we estimate that the Kerr coefficient is $\gamma \sim 2$–3 $W^{-1} \, km^{-1}$. We have chosen $\gamma = 2 \, W^{-1} \, km^{-1}$ in the numerical simulations displayed in Fig. 4. The PS displayed in Fig. 3h corresponds to $\gamma = 3 \, W^{-1} \, km^{-1}$. All numerical integrations are performed using an adaptive stepsize pseudospectral method, using a mesh of 2048 or 8192 points, over a temporal window of $\Delta T = 250 \, ps$.

In numerical simulations presented in this letter, we neglect linear losses (≃0.5 dB) and stimulated Raman scattering. These approximations provide precise and quantitative agreement between experiments and numerical simulations at moderate powers (<2 W, see also Supplementary Note 3 and Supplementary Fig. 5). Additional numerical simulations show that stimulated Raman scattering has to be taken into account in order to reproduce very precisely the experimental PDFs and optical spectra at high values of the mean power (that is, $\langle P \rangle = 4 \, W$). However the main physical results (formation of RWs, emergence of breather-like structures and heavy-tailed PDFs) are not affected by stimulated Raman scattering. Note finally that we consider here the deterministic 1D-NLSE (without any additional noise occurring along the propagation inside the fibre).

The random complex field $\psi(t, z = 0)$ used as initial condition in numerical simulations is made from a discrete sum of Fourier components:

$$\psi(z=0,t) = \sum_m \widehat{X_m} e^{im\Delta\omega t}. \quad (2)$$

with $\widehat{X_m} = 1/\Delta T \int_0^{\Delta T} \psi(z=0,t)e^{-im\Delta\omega t}dt$ and $\Delta\omega = 2\pi/\Delta T$. The Fourier modes $\widehat{X_m} = |\widehat{X_m}|e^{i\phi_m}$ are complex variables. We have used the so-called random phase (RP) model in which only the phases $\phi_m$ of the Fourier modes are considered as being random[41]. In this model, the phase of each Fourier mode is randomly and uniformly distributed between $-\pi$ and $\pi$. Moreover, the phases of separate Fourier modes are not correlated so that $\langle e^{i\phi_n}e^{i\phi_m}\rangle = \delta_{nm}$ where $\delta_{nm}$ is the Kronecker symbol ($\delta_{nm} = 0$ if $n \neq m$ and $\delta_{nm} = 1$ if $n = m$). With the assumptions of the RP model above described, the statistics of the initial field is stationary, which means that all statistical moments of the complex field $\psi(z = 0, t)$ do not depend on $x$[41]. In the RP model, the power spectrum $n_0(\omega)$ of the random field $\psi(z = 0, t)$ reads as:

$$\langle \widehat{X_n}\widehat{X_m}\rangle = n_{0n}\delta_{nm} = n_0(\omega_n) \quad (3)$$

with $\omega_n = n\Delta\omega$. In our simulations, we have taken a random complex field $\psi(z = 0, t)$ having a Gaussian optical power spectrum that reads

$$n_0(\omega) = n_0\exp\left[-\left(\frac{\omega^2}{\Delta\omega^2}\right)\right] \quad (4)$$

where $\Delta\omega = 2\pi\Delta\nu$ is the half width at $1/e$ of the power spectrum. Statistical properties of the random wave have been computed from Monte Carlo simulation made with an ensemble of 10$^4$ realizations of the random initial condition.

**Data availability.** The data that support the findings of this study are available from the corresponding author on request.

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

## Acknowledgements

This work has been partially supported by the Agence Nationale de la Recherche through the LABEX CEMPI project (ANR-11-LABX-0007) and the OPTIROC project (ANR-12-BS04-0011 OPTIROC), and the EQUIPEX FLUX (ANR-11-EQPX-0017) as well as by Lille 1 University (BQR Émergence-Innovation) and by the Ministry of Higher Education and Research, Nord-Pas de Calais Regional Council and European Regional Development Fund (ERDF) through the Contrat de Projets Etat-Région (CPER Photonics for Society P4S). The authors acknowledge Miguel Onorato (Torino, Italy), Gennady El (Loughborough, UK), Tamara Grava (Trieste, Italy) and Goery Genty (Tampere University, Finland) for fruitful discussions. The authors are grateful to Francois Anquez and the Biophysics of Cellular Stress Response group of the PhLAM for the fruitful discussions, their crucial help, and for providing the sCMOS Camera. The authors are also grateful to Arnaud Mussot, Rémi Habert and the photonics group of the PhLAM for fruitful discussions, for the equipments (the ps laser), and for the measurement of the GVD of the fibre. The authors thank Nunzia Savoia for the everyday work on the femto laser and Marc Le Parquier for his crucial contribution in the development of the time lens.

## Author contributions

P.S. and S.B. have initiated the work. The design and realization of the experimental setup on rogue wave generation has been performed by S.R., P.S. and R.E.K. The time-microscope has been designed and realized by C.S., C.E., S.B. and P.S. All authors participated to data acquisition that has been essentially performed by R.E.K. Data analysis has been performed by A.T. and P.S. Numerical simulations and code development have been performed by S.R., P.S. and C.E. P.S., S.B., S.R., C.S., C.E. have written and revised the manuscript.

## Additional information

**Competing financial interests:** The authors declare no competing financial interests.

