## [Peer Review File · Nature Communications]

Reviewer #1 (Remarks to the Author):

Application of a so-called time microscope is discussed to supercontinuum-seeded solitons that are generated in a nonlinear fiber. This setup provides real-time temporal resolution of picosecond transients, similar to an optical oscilloscope. The setup is operated at 500 Hz repetition rate and has been used to record a large number of transients, which enables meaningful statistical evaluation. This evaluation supports the claim that fiber solitons, Akhmediev breathers etc. underlie "rogue-wave" statistics, whereas simple linear propagation does not. Unfortunately, the authors do not quite seem to realize that their data shows a quite meaningful difference between the perfectly normal-distributed input to their nonlinear fiber and the heavy-tail distribution at the output, see my detailed remarks below. So the statistical analysis is not quite complete and satisfactory yet. Moreover, I found some parts of the introduction and abstract fairly confusing, in particular concerning the connection of the experiments to ocean rogue waves. I think that the authors really need to be a bit more careful here.

1. The first two sentences in the abstract directly imply that rogue waves are, in general, coherent structures. This is certainly true for the solitons under investigation, but looking into published ocean rogue wave records, this statement appears highly questionable, see, e.g., the five examples published as Fig. 1.4 in the book "Rogue waves in the ocean" by Kharif, Pelinovsky, and Slunyaev. The statement also does not hold for other optical experiments, e.g., Arrechi et al., PRL 106, 153901 (2011), who actually showed that the very lack of coherence may also lead to rogue waves. Solitons are coherent and some but certainly not all rogue waves are. Coherence therefore does not appear a good criterion.

2. Let me further state that the assumption of ocean rogue waves being solitons has recently been challenged by several researchers; ocean rogue waves may simply arise from linear random interference of a large number of waves. I think the authors need to be more careful here with the frequent use of the word "rogue wave". They actually take it as a given what they eventually prove, namely that Akhmediev breather solitons underlie non-Gaussian statistics. If you call them rogue wave right from the start you are already anticipating the result of the study.

3. It is then further claimed that spatio-temporally isolated optical rogue waves have never been observed before. This is not true either, see, e.g., Fig. 5 in PRL 111, 243903 (2013). Even though the physics of the rogue waves was different, essentially the same trick was used in the latter PRL, i.e., going to about 1 kHz repetition rate and using a fast camera for recording the events.

4. First sentence in the introduction: "Common ocean waves are weakly nonlinear random objects having nearly Gaussian statistics..." If this were true then there would be no rogue waves! Ocean wave heights actually have a near-Rayleigh distribution, so there is a heavy tail in the distribution. The "weakly nonlinear" also shows that the authors do not quite know what they are talking about. Ocean waves are actually not sinusoidal but require a so-called Stokes correction, which, to leading order, lifts the crests as much as the troughs. Obviously, this does not affect the Rayleigh-distributed crest-to-trough wave height, but one cannot safely state that ocean waves are weakly nonlinear objects either.

5. Akhmediev breathers are certainly considered by some people as the prototypical rogue wave, but the main supporter of this theory is Nail Akhmediev himself, and one can certainly simply choose to believe him, but it may also turn out that he is wrong. But it seems to me that that the authors have all the data to verify the statement rather than choosing to simply believe in what Akhmediev says.

6. The rogue waves previously observed during nonlinear fiber propagation by Solli et al. were almost certainly no Akhmediev breathers, but rather boring fundamental solitons. I think it is important to say this here, because otherwise the question could arise what sets the current study apart from the Solli experiment. Then this work and by many other optical rogue wave experiments are called "indirect evidence" for rogue waves. I think this is extremely unfair. And unnecessary!

7. Most researchers in the ocean community seem to believe that the real rogue waves have nothing

to do with solitons, and some of them are quite upset if the word "rogue wave" is used as a selling vehicle in optics. I counted 54 appearances of "rogue wave" or "RW" in the current manuscript. Maybe we can agree to get this number down to a handful and to mainly talk about Akhmediev breathers or solitons here while initially pointing out that a connection has been suggested, but that there is currently no statistical evidence that allows making this connection? Maybe we can also remove "rogue wave" from the title?

8. The PDF in Fig. 3: To be anywhere fair in the frequently implied comparison with ocean waves, one should show the field profiles rather than power profiles and also apply the sqrt operation to the abscissa of Fig. 3(a). With the commonly used threshold of 4 times the standard deviation, this would only make waves above $16\langle P \rangle$ actual rogue waves, i.e., 3e and f are not, but Figs. 3d and h certainly are. Maybe the authors did not quite see the problem, but rogue waves are supposed to be rare. In the current representation, non-rogue waves actually seem to be the rare ones, even the initial condition already contains tons of rogue waves. The PDF of the linear case should be a parabola in the log representation of Figs. 3(a) and 4(a) according to theory. This case is known as linear interference of long-crested waves in ocean science. Then it would also be helpful to mark the rogue threshold at $4\langle E \rangle$ or $16\langle P \rangle$ in these plots. It's all beautifully there, you just have to plot the correct statistics.

9. The authors say that one-dimensional deep-water waves are ruled by the NLSE. I am not sure about the "deep-water" aspect, but yes, 1D solitons have first been discovered in a canal near Edinburgh, maybe a very deep canal, but what does this have to do with the ocean, the surface of which is undoubtedly two-dimensional? And 2D solitons are unstable. There is another modulation instability.

In summary, I think that these are beautiful experiments that should definitely be published. However, the authors should not make the mistake of automatically equating solitons and rogue waves. I think that in particular the introductory paragraph needs to be completely rewritten and that the frequency of "rogue wave" should be brought down to an acceptable 5 or 6 in the manuscript. Then I believe that the authors should refrain from making any statements on ocean physics. This is clearly not their area of expertise, and an expert can immediately sense this in each of their attempted sentences on the subject. Finally, Figures 3 and 4 are clearly missing their marks. It's all about field amplitudes, not intensities, or everything will be rogue.

Provided that all these mandatory changes are being implemented, I would lean towards recommending acceptance afterwards, but the introduction currently appears absolutely unacceptable. I think that I offer the authors a golden bridge here how they can turn the introduction around to make this paper worth publication in Nature Communications.

Reviewer #2 (Remarks to the Author):

The work under review is related to real time measurements of rogue waves as unique events.

The genuine discovery made in this work is the direct observation of breather-like rogue waves generated in an optical fibre. From the theoretical point of view, these are breathers described by the nonlinear Schroedinger equation. The latter are new objects in physics in contrast to well known solitons.

In optics, the use of direct methods to see them have not been possible in the past. The common methods such as the autocorrelation technique did not allow to see the actual shape of the short pulses. For, example, it did not allow to measure asymmetry of the pulses. The snapshots in Figs. 1, 3 and 4 (particularly 3h and 4g) are really remarkable in this sense. These snapshots allowed the authors to make the direct comparison of experimental data with the theoretical curves.

The measurements of real shapes of isolated rogue waves are done for the first time. The concept of

time microscopy used in this work can be considered as a revolution in observation of rogue waves. Transferring the profile in time to a spatial profile made it possible to find the real time profiles of the pulses of interest. Importantly, this is done when operating with a chaotic optical string.

Especially impressive is the fact of using random initial conditions. In all previous cases, experimentalists had to deal with periodic signals as the equipment has been mostly designed for them. Detecting individual pulses here is a significant progress. High dynamical range of the time microscope designed by the authors is another achievement that made it possible to register the rogue waves.

This work is of pioneering nature and it opens new possibilities in ultra-short pulse optics. The paper is clearly written, well illustrated and provides references to all essential previous publications. I liked the simplicity and clarity of explanations that would allow other researchers to repeat the measurements and to make further steps in this emerging field of research. Moreover, the text is carefully edited and does not contain typos which is a common problem in other works that I was reviewing recently. I did not find a single one, anyway.

I do not hesitate to recommend this work for publication in Nature Communications in its present form.

Reviewer #3 (Remarks to the Author):

Ref. report on Manuscript 91954

Title: Direct observation of Rogue Waves in optical turbulence using Time Microscopy

Authors: P Suret, et al.

This manuscript shows a measurement of very short pulses of light after propagation through an optical fibre. The technique used to measure picosecond pulses in real time is called "time microscopy" which is already known since 2008 (R. Salem et al. Opt. Lett. 33, 1047 (2008)). This measurement puts into evidence the existence of high intensity pulses which are much higher than the average whenever the average input power exceeds a given threshold. It is already known that the propagation in optical fibres is able to generate the so called optical rogue waves. References are already given in this manuscript. However this work seems to be the first time a direct measurement have been done in this type of system but it is not the first time that optical rogue waves have been directly detected in an optical system (see for example A. Hnilo et al. Opt. Lett, Nov 2011 in mode locked system, Bonatto et al., PRL, July 2011 in laser with optical injection, and Ref.3 in this manuscript includes several other references)

Thus considering that extreme events or optical rogue waves have been detected in optical systems, that optical rogue waves have been "observed" in this type of system, and finally that the technique used here is already known, I think that the manuscript does not have enough significant results to justify publication in Nature Communications.

On the other hand I can give a detail analysis of the different parts of the manuscript and the results presented here:

1) Some general statements in the introduction relating rogue waves observed in the ocean with extreme events in optical systems are in my opinion too much speculative. I think that equivalence between two very complex dynamical systems can be established only through measurements that are not available in the comparison between the ocean and optical systems. Templates, bifurcation

diagrams and other techniques can not be available. Therefore I prefer that the reader will not imagine that optical rogue waves are the same as rogue waves in the ocean.

2) The manuscript says that "common ocean waves are weakly nonlinear random objects having nearly Gaussian statistics". I suppose that researchers in oceanography will never accept such statement. In fact most of them tried to developed models that will not give Gaussian statistics because Gaussian models will give perfect symmetric waves which clearly are not observed very often in a "moving" sea!!

3) An oceanic rogue wave is not necessarily a spatially localised peak. The definition of a rogue waves for people working on oceanography is based only on the high of the wave compared to the average. There is no mention about the propagation distance. I do not think in optics it requires such localisation. On the other hand the measurement presented here is only in time and not in space because the system is by definition a 1D system. The spatial coordinate being the propagation one is directly related to time, then I do not understand. What is the meaning of localisation? A short pulse in time is essentially the same as localisation in the spatial coordinate.

4) The amplification of a pulse giving rise to high intensity is not surprising in a nonlinear system like this one.

5) If I understood correctly random pulses in time at very well defined optical frequency are propagated through the optical fibre. The output shows extreme events. However the manuscript is saying that the system is turbulent. There is no proof at all all along the manuscript that the system is turbulent. What definition of turbulence have been used? What measurements indicate or put in evidence the turbulent character of the dynamical behavior of the system? I did not see any measurement of the loss of spacetime correlation or an inverse cascade in the spectrum? It seems to me that such evidence is not presented here nor in the experimental results or even in the numerical ones.

6) In page 5 the manuscript reads: "The emergence of coherent structures is a general mysterious feature of stochastically driven processes....." I did not get such statement. The generation of coherent structures from noise is a general behaviour in many physical systems, being lasers one of the most well known examples in optics where a coherent beam grows from spontaneous emission.

7) Finally I do not understand the comparison with the NLSE. If the NLSE describes the system and the system is driven by a random source, then it is not enough to put a random initial condition but it requires a noise term included into the equation. There is a fundamental difference in dynamics between the meaning of a noise driven system with respect to a random initial condition even if the system is conservative.

In conclusion I do not recommend publication in Nature Communications. Probably the paper with some modifications could be appropriate for another physical journal.

Answer to Reviewer #1 :

We thank Reviewer 1 for its extremely positive comments about the importance of our experimental results. Reviewer 1 makes some important recommendations about our introduction. Moreover Reviewer 1 emphasizes that we should better stress conclusions about the link between statistics and the observation of the structures. We fully agree that this conclusion is the most important of our work.

We understand from the comments of the reviewer 1 that the initial presentation of the context of our work may seem unclear to some readers. In particular, a few sentences about the link between hydrodynamics and optics may be misunderstood. The mechanisms of emergence of rogue waves (RW) both in hydrodynamics and in optics are not unique and they depend on the exact nature of experiments under consideration. In other words, one cannot say that there is ONE kind of RW. For example, as pointed out by the reviewer 1, if one considers the definition of RW based on a threshold ($H > 2.2 H_s$), RW might arise both from a trivial linear superimposition of waves or from some nonlinear interactions of waves.

We have referred to ocean waves in the introduction to situate RWs from a historical perspective but we do not claim that the structures observed in our experiments can be associated to oceanic RW. From the comments of reviewer 1, we understand that the perspective of our work has to be more precisely explained. Our experiments are very close in their principle to the hydrodynamical experiments performed by Onorato et al. (Phys. Rev. E 70, 067302, 2004). However, note that in the optical experiments presented in our manuscript, the nonlinearity is very large as compared to those hydrodynamical experiments (Wave breaking is a non-negligible mechanism that prevent to explore the propagation of water waves in strongly nonlinear regimes characterized by high steepness.)

The fundamental context of our study is the so-called “integrable turbulence” introduced recently by Zakharov. In order to clarify the context of our work in full accordance with the remarks of the Reviewer 1, we have fully re-written a more detailed introduction. We have also changed the abstract. We have clarified several points in the core of the article and we have added a supplementary material (Pdf of $|\psi|$ and threshold of RW). We have also deeply changed the conclusion of the article.

Moreover, we have used the word OPTICAL rogue waves in the title and we refer

explicitly to the context of integrable turbulence. The new title reads :”Single-shot observation of Optical Rogue Waves in integrable turbulence using Time Microscopy”.

/////

Detailed Answers to comments/questions of Reviewer 1

////////////////

General Reviewer 1 comment : “ The setup is operated at 500 Hz repetition rate and has been used to record a large number of transients, which enables meaningful statistical evaluation. This evaluation supports the claim that fiber solitons, Akhmediev breathers etc. underlie "rogue-wave" statistics, whereas simple linear propagation does not. Unfortunately, the authors do not quite seem to realize that their data shows a quite meaningful difference between the perfectly normal-distributed input to their nonlinear fiber and the heavy-tail distribution at the output, see my detailed remarks below. “

////////////////

We agree on the main point : our study demonstrates for the first time a link between the emergence of heavy tailed statistics and breather-like structures. We are perfectly aware of the importance of the statistical changes arising from nonlinear propagation. In our original manuscript, this part was unfortunately reduced to a small paragraph and the important idea that our results correlate these statistical changes to the emergence of coherent structures was insufficiently well highlighted.

The experimental demonstration in optics of heavy-tailed deviations from gaussian statistics is in itself not new (we published it in PRL (2015), ref 38 of our manuscript) but the measurement of PDF was made with an optical sampling technique that did not provide the real time observation of the underlying dynamics. The fact that we observe the same statistical behavior with the time microscope as the one already observed with the optical sampling method is of a fundamental importance because it demonstrates the link between the heavy tailed PDF and the emergence of the coherent structures, in particular those that are similar to breathers solutions of 1D-NLSE. We have emphasized this point in the introduction, in the core of the article and also in the conclusions (see below)

////////////////

Question 1 of the Reviewer 1 : “ The first two sentences in the abstract directly imply that rogue waves are, in general, coherent structures. This is certainly true for the solitons under investigation, but looking into published ocean rogue wave records, this statement appears highly questionable, see, e.g., the five examples published as Fig. 1.4 in the book "Rogue waves in the ocean" by Kharif, Pelinovsky, and Slunyaev. The statement also does not hold for other optical experiments, e.g., Arrechi et al., PRL 106, 153901 (2011), who actually showed that the very lack of coherence may also lead to rogue waves. Solitons are coherent and some but certainly not all rogue waves are. Coherence therefore does not appear a good criterion.”

////////////////

We fully agree with Reviewer 1 and it was not our intention to mention that ALL the

Rws are solitons or some other coherent structures. As we explained above, our work enters within the fundamental framework of integrable turbulence where some numerical predictions of the emergence of coherent structures have been made. We have completely changed the abstract and the introduction in order to avoid any possible confusion about this point. We have in particular added many references to different points of view on RWs.

//////////

2. Reviewer 1 “Let me further state that the assumption of ocean rogue waves being solitons has recently been challenged by several researchers; ocean rogue waves may simply arise from linear random interference of a large number of waves. I think the authors need to be more careful here with the frequent use of the word "rogue wave". They actually take it as a given what they eventually prove, namely that Akhmediev breather solitons underlie non-Gaussian statistics. If you call them rogue wave right from the start you are already anticipating the result of the study.”

//////////

Again, we agree with the Reviewer 1. There are many active debates about the mechanisms of emergence of RWs. We do think that these mechanisms are not unique and that the emergence of solitons on finite background such as Akhmediev breathers represents one possible scenario in some specific cases. In particular, the wave system has to be one-dimensional and it has to be described by the focusing 1D NLS equation at leading order. In hydrodynamics, this corresponds to water waves with narrow spectrum propagating in one-dimensional tanks in the so-called deep-water case (see book of Kharif, the book of Osborne...). In ocean, it is an open question to determine whether long crested waves with narrow directional spreading may lead to this kind of Physics (see the book of Kharif).

In laboratory experiments made in a 1D water tank, Onorato et al have shown in 2004 that the nonlinear propagation of random waves characterized by gaussian statistics of the surface elevation at initial stage produces non gaussian statistics : the precise modeling of these experiments is still an open question (NLS vs Dysthe equation in particular).

In optical fibers, the question is much more simple because it is possible to design experiments that are very well described by 1DNLSE (this corresponds to experiments presented in our manuscript). In this context, our experiments demonstrate that breathers-like structures emerge from the nonlinear propagation of random waves (in particular we demonstrate the emergence of the Peregrine soliton). This represents our main result and it was already emphasized in the initial manuscript (last sentence of the abstract, and the paragraph starting with : “In the last years, the common and shared conjecture is that breather-like solutions of 1D-NLSE such as PS or Akhmediev breathers represent prototypes of RWs [11, 14-16, 18-20]...”

Taking into account the remark of the Reviewer 1, we have completely re-written the introduction in order to clarify the conclusions of our work. In particular, we now write :

- “In all these experiments, the so-called optical RWs (and more generally extreme events) correspond to a large variety of phenomena, often not comparable to ocean waves, and their generation cannot be ascribed to a unique physical mechanism [1,

3, 16–18] “

- “It is conjectured by some authors that breather solutions of the 1D-NLSE such as the Peregrine soliton (PS) may represent prototypes of RWs [20, 22, 24, 26–28]”
- “The questions related to the local emergence of coherent structures compatible with breathers solutions of the 1D-NLSE are still completely open in the context where waves randomly fluctuate in space or time. The problem of random (or partially coherent) inputs in wave systems described by integrable equations such as 1D-NLSE enters within the fundamental framework of the so-called “integrable turbulence”, a new research field recently introduced by V. E. Zakharov [34–39]”

Moreover, in the revised version of the manuscript, we have taken into account the remark of the Reviewer 1 : we understand that the too frequent use of the word “Rogue Waves” may be confusing. In our paper, we bring a link between the emergence of breathers in integrable turbulence and heavy tailed PDFs. We do not want to infer that formation of breather-like structures is the only possible scenario for the formation of Rws. Moreover, we do not draw conclusions about RW in Oceans. We have modified the conclusion of the article to avoid any misunderstanding. In particular one of the last paragraphs now reads :

“The conclusions which are drawn here in optical fibre experiments well described by the 1D-NLSE cannot be directly extrapolated to phenomena found in the oceans. Let us emphasize however, that the principle of our experiment is identical to the one of some previous experiments performed in one-dimensional deep water tank [54]. Starting from random initial conditions, those hydrodynamical experiments have also demonstrated the formation of heavy-tailed statistics. Note finally that the extreme amplitudes observed in our optical fibres experiments could not be observed in water waves that are strongly limited by the wave breaking phenomenon [1, 55, 56].”

//////////

Question 3. Reviewer 1 : “It is then further claimed that spatio-temporally isolated optical rogue waves have never been observed before. This is not true either, see, e.g., Fig. 5 in PRL 111, 243903 (2013). Even though the physics of the rogue waves was different, essentially the same trick was used in the latter PRL, i.e., going to about 1 kHz repetition rate and using a fast camera for recording the events.”

////

We do agree with the reviewer. Once again the presentation of our initial manuscript was confusing and we did not want to claim that isolated optical rogue waves have never been observed before in ANY system. However, our experiments provide the first real-time observation of temporal RW objects in the context of nonlinear random waves rapidly fluctuating in time (at the ps time scale). This picosecond time scale typically characterizes solitons or breathers in optical fiber experiments.

The experiments presented in the paper mentioned by the Reviewer are very nice and interesting but it is important to emphasize that the “trick” is not the same. We use a camera at 500Hz repetition rate but this is the last step of a device composed of several

stages that transform temporal evolution to spatial coordinate. In PRL 111, 243903 (2013), the authors investigate SPATIO-temporal dynamics of 1D(propagation)+2D (transverse profile) systems in the context of optical filamentation with a low temporal resolution. The camera is used to directly observe the transverse profile of a beam. In our case, we resolve the sub picosecond dynamics of a 1D (propagation) system described by the 1D NLSE and our camera is not used for the characterization of a beam profile but to record a signal that is a “picture” of the temporal dynamics.

As a conclusion, one have to distinguish spatial experiments from temporal experiments. We have clarified this point in the new version of our article and we now cite the paper mentioned by the Reviewer 1 and also another one about spatial experiments.

//////////

4. Reviewer 1 : First sentence in the introduction: "Common ocean waves are weakly nonlinear random objects having nearly Gaussian statistics..." If this were true then there would be no rogue waves! Ocean wave heights actually have a near-Rayleigh distribution, so there is a heavy tail in the distribution.

//////

In order to avoid any confusion, we have removed this first sentence.

However, let us clarify an important point here : the Ocean is indeed weakly nonlinear ON AVERAGE : the steepness cannot exceed 0.1 because of the wave breaking phenomenon (ref. 55 if our manuscript).

We refer to the Gaussian statistics of surface elevation and Reviewer 1 refers to the Rayleigh distribution of wave height. We agree with Reviewer 1 that we have to clarify this point. There is a lot of confusion in the optical literature about the nature of the statistical distributions that characterize all the variables which might be measured. However in the books about Oceanography (for example in the section 2.2.1 of the book of Kharif cited by Reviewer 1), the following facts are clearly explained :

- considering the surface elevation η with a Gaussian statistics and a narrow band approximation one can write $\eta = |\psi| \cos(k \cdot r - \omega t + \phi)$.
- $|\psi|$ is RAYLEIGH-DISTRIBUTED
- the wave height H (in the narrow band approximation) is also Rayleigh-distributed
- the exceedance probability of wave height $> H$ is a Gaussian
- The probability distribution of the $|\psi|^2$ that is relevant in optics is the exponential distribution.

This relationship between statistics holds in 1D problem (see the book of Optical statistics of Goodman for example, ref. 51 in the manuscript)

We emphasize that in the literature, saying that a field has a statistics characterized by a “heavy-tailed distribution” generally means that the field has a PDF that exhibits tails that are larger than those given by the central limit theorem.

In hydrodynamics, the central limit theorem has to be applied on the surface elevation η that is physically constructed from the superposition of numerous Fourier components. Thus, in the context of ocean waves and more generally of random waves, one has to compare :

- the statistics of η with the Gaussian distribution
- the statistics of $|\psi|$ with the Rayleigh distribution
- the statistics of $|\psi|^2$ with the exponential distribution.

In our experiments, the random electric field used as initial condition is made from a linear superposition of numerous Fourier components and it obeys the central limit theorem. We examine the question of statistical changes arising from the nonlinear propagation of these random waves that have gaussian statistics at initial stage. The variable chosen for examining this question can be either $|\psi|$ or $|\psi|^2$ (even though the most natural one in optics is the power $|\psi|^2$).

We thought this point was clear in the original manuscript (see the sentence “probability density function (PDF) of the optical power emitted by the ASE source is systematically very close to the exponential distribution that corresponds to a Gaussian statistics for the field. On the contrary, the PDF of light power at the output of the nonlinear fibre is found to exhibit heavy-tailed deviations from the exponential distribution, thus confirming the generation of RWs.”)

We understand from the remark of the Reviewer 1 that we have to be more pedagogical. We thus explicitly refer to the central limit theorem and refer to the books of Kharif et al. and of Goodman in order to avoid any confusion. Moreover we have added a section in the supplementary material to clearly relate all the possible PDFs to their corresponding variable.

/////

Reviewer 1 says : The "weakly nonlinear" also shows that the authors do not quite know what they are talking about. Ocean waves are actually not sinusoidal but require a so-called Stokes correction, which, to leading order, lifts the crests as much as the troughs. Obviously, this does not affect the Rayleigh-distributed crest-to-trough wave height, but one cannot safely state that ocean waves are weakly nonlinear objects either.

/////

We do agree with this statement and we are fully aware of the Stokes waves. But we do insist about the weak nonlinearity of the Ocean (on average) : the estimation of the average steepness (that measures the strength of the nonlinearity) in the ocean is typically 0.1 (see the Ref. 55). The main reason is the existence of the wave breaking. These statements are true for deep water and narrow band approximation (the second order nonlinearity leading to the Stokes Waves is then non resonant) and do not mean that nonlinearities do not play crucial role in the oceans.

We want to insist that this discussion was not a key point of our introduction : on the contrary, our work is performed in a nonlinear regime. We thus remove any ambiguous statement about ocean in the introduction but we explain in the conclusions that RW as

high as in our experiments cannot exist on the surface of the Ocean because of wave breaking.

//////////

Point 5. of the Reviewer 1 : Akhmediev breathers are certainly considered by some people as the prototypical rogue wave, but the main supporter of this theory is Nail Akhmediev himself, and one can certainly simply choose to believe him, but it may also turn out that he is wrong. But it seems to me that that the authors have all the data to verify the statement rather than choosing to simply believe in what Akhmediev says.

//////////

Nowadays, the question of the identification of RW is one of the most important question of the field. We agree that there is a very active debate in the recent literature about the prototypes of RWs. Tens of papers -also in hydrodynamics- support the claim of Akhmediev (see for example ref 23 of the manuscript or also Physica Scripta, "Note on Breather Type Solutions of the NLS as Models for Freak-Waves" Dysthe and Trulsen).

In our introduction, we want to emphasize that this conjecture is supported by a large part of the RW community (even if it is not by every body). And our perspective is exactly along the lines suggested by Reviewer 1 : our main goal is to provide some data in order to test the validity of this conjecture (at least in the context of integrable turbulence). We understand from the interesting remarks of the Reviewer 1 that this was unclear in the original question of our manuscript. We have thus stressed out this point in the introduction of the revised manuscript. Note finally that we do not want to enter within the debate about the extremely complex Physics of Ocean. We "just" provide experimental data which show that solutions of 1D can emerge in nonlinear random waves.

Reviewer 1 might have missed some part of our main conclusion : our data analysis seems to show that the Peregrine soliton frequently appears at the output of the optical fiber. We show this in Fig. 3.h by superimposing the exact analytical shape of the Peregrine soliton. We want to emphasize that here, the important adjustable parameter is the value taken by the background. We relate this behavior to a recent mathematical theorem (Bertola and Tovbis, 2003) about nonlinear propagation of a single hump : it demonstrates that the Peregrine emerges at the gradient catastrophe point in the semi classical limit. The background is not the averaged power as expected from modulation instability of a plane wave (as in theoretical ref. 28 of our manuscript). The background has a value that is exactly the maximum power at the gradient catastrophe point.

Our experiments thus demonstrate that (at least in integrable turbulence), some of the breather solutions can emerge from nonlinear propagation of random waves. However, we consider that it is difficult to draw some other conclusions from our data. The first reason is that the Akhmediev breather is a periodic solution in time on an infinite line. As we consider random waves, Akhmediev breather cannot appear at all time. One can only consider a structure that looks like a breather **locally**. A systematic data analysis on random waves is a mathematical complex open question : as far as we know there is no established procedure to detect a structures "close to" a known solution in a turbulent flow.

"By eyes", we may think that isolated breathers compatible with Peregrine or Kuznetsov-Ma solitons seem more probable than Akhmediev breather. But we think that further investigations are needed. The typical example of the Fig. 3.c shows also that

several structures often co-exist. There is no criteria to decide whether these structures are simple “solitons” (with zero boundary conditions) or deformation of an Akhmediev breather ? As we wrote in the initial conclusion, the next bottleneck in the analysis of coherent structures emerging from random backgrounds is the phase measurement. If one wants to draw some definite conclusions about the nature of these breather-like structures, phase measurement is needed. To the best of our knowledge, the phase measurement of random waves with sub-picosecond time scale is an experimental open and extremely challenging problem.

This question of the Reviewer 1 gives us the opportunity to improve the discussion about the experimental results. In particular we have commented more in details the Fig. 3. (In particular we add : “It is difficult to draw clear conclusions from single-shot observation displayed on Fig.3.(c) : these structures might be compatible with interacting fundamental solitons or with randomly perturbed periodic Akhmediev breathers.”)

//////////

6. The rogue waves previously observed during nonlinear fiber propagation by Solli et al. were almost certainly no Akhmediev breathers, but rather boring fundamental solitons. I think it is important to say this here, because otherwise the question could arise what sets the current study apart from the Solli experiment. Then this work and by many other optical rogue wave experiments are called "indirect evidence" for rogue waves. I think this is extremely unfair. And unnecessary!

//////////

We agree with the referee 1 that there are fundamental differences between our work and the work of Solli :

- our experiments are well described by the focusing and integrable 1DNLSE whereas (supercontinuum) experiments of Solli et al can only be described in the framework of a generalized NLSE that is not integrable. Therefore, strictly speaking, pure fundamental solitons cannot emerge in the experiments of Solli et al. However, it is true that the emission of solitons in supercontinuum is a general accepted mechanism underlying supercontinuum generation.

Therefore, we would find that it would be confusing to draw rigorous conclusions about the soliton nature of coherent structures found in the work made by Solli et al.

- the field launched inside the fiber in Solli et al. (2007) is a coherent pulse (involving thus zero boundary conditions) whereas we launch partially coherent waves (thus corresponding to non zero boundary conditions in theory). In our case, both fundamental solitons and solitons on finite background can co-exist.

Moreover, the most important point is the technique of measurement : in the experiments made by Solli et al. and in all the next studies made with optical fibers, no DIRECT (single shot) observation of structures has been reported. In the paper by Solli et al, the authors first filter the signal in the Fourier spectrum by using an optical filter so that both the statistical and dynamical informations that are obtained correspond only to a

limited part of the Fourier spectrum of the optical wave. In literature devoted to fast **temporal** RW (mainly optical fibers experiments), there are two kind of works :

- indirect observation by using optical filter (paper of Solli et al., ref. 4 in the manuscript)
- observation with photodiodes having detection bandwidth much narrower than the bandwidth of the optical signal.

In both cases, the observation of RW is not performed “in real time” because the bandwidth of detection is extremely limited. It is very important (and fair) to emphasize this point. However, we understand from the comment of the reviewer 1 that the word “direct observation” can be misunderstood here. We have replaced “direct” by “single-shot” in the title).

Note moreover, that some papers use the word “optical rogue waves” in context far from the initial context (link between optics and hydrodynamics). In particular, self pulsing dynamics (with slow time scale) in lasers is sometimes called “RW”. Even if those works are interesting, the physical systems under consideration are not described the generic 1D-NLSE physics. We understand that it is important to distinguish between the experiments devoted to 1D NLSE Physics and other devices, in particular dissipative systems such as lasers.

In order to clarify all these points, we have added the following paragraph :

“From the technical point of view, it is important to note that, up to now and despite the numerous experiments devoted to optical RWs, the precise time domain observation of coherent structures compatible with 1D-NLSE solutions and the study of their shape were not possible. Contrary to the direct observation of spatial structures in experiments in which the intensity profile of a light beam is recorded with a camera [9, 45, 46] the fast time scales of fluctuations (picoseconds or less) involved in single-mode fibre experiments make single-shot recording of RWs a particularly challenging task. Pioneer works performed in optical fibres hence did not provide single shot observation of RW objects but evidence of it by using e.g. spectral filtering [4, 6, 47] or statistical measurement from optical sampling techniques [38]. In the case of mode-locked laser dynamics, statistical measurement of the pulse amplitude can also been performed with standard photodetector [12]. However, up to now, in all these experiments, the real time and single-shot observation of the shape of the picosecond time scale structures generated from the nonlinear propagation of random waves has never been reported.”

//////////

7. Most researchers in the ocean community seem to believe that the real rogue waves have nothing to do with solitons, and some of them are quite upset if the word "rogue wave" is used as a selling vehicle in optics. I counted 54 appearances of "rogue wave" or "RW" in the current manuscript. Maybe we can agree to get this number down to a handful and to mainly talk about Akhmediev breathers or solitons here while initially pointing out that a connection has been suggested, but that there is currently no statistical evidence that allows making this connection? Maybe we can also remove "rogue wave" from the title?

///

We partly agree with the Reviewer 1 : the word RW is now widely used in the optical community to refer only to “high amplitude events” (even without any link with hydrodynamics). However, our experiments performed in a 1D focusing NLS system are analogous to experiments performed in 1D water tank with random initial waves (see Onorato et al 2004). The main difference is that our experiments deal with strongly nonlinear whereas wave breaking prevent the occurrence of strong heavy tailed statistics with gravity waves. We add “optical” before “rogue waves” in the title in order to avoid possible confusion with “oceanic” RWs. We also agree that in the text, it is clearer to use several times “coherent structures” instead of the unique word “RW”. We thus follow the proposition of Reviewer 1 to decrease the number of references to the word RW.

However, we estimate that it is important to keep the word RW in the title, in the abstract and several times in the text. We use now the full phrase “Optical RW” and moreover, we say explicitly in the manuscript that the optical phenomena are not identical to the ones observed in Ocean (in particular in the conclusion : “The conclusions which are drawn here in optical fibre experiments well described by the 1D-NLSE cannot be directly extrapolated to phenomena found in the oceans. [...] Note finally that the extreme amplitudes observed in our optical fibres experiments could not be observed in water waves that are strongly limited by the wave breaking phenomenon”). Moreover a large part of the optical RW community will nowadays find strange to avoid this phrase in our paper.

Finally, we want to emphasize again that our study is devoted to Optical Rogue waves in Integrable turbulence and that we have changed the title accordingly.

/////

8. The PDF in Fig. 3: To be anywhere fair in the frequently implied comparison with ocean waves, one should show the field profiles rather than power profiles and also apply the sqrt operation to the abscissa of Fig. 3(a). With the commonly used threshold of 4 times the standard deviation, this would only make waves above 16 actual rogue waves, i.e., 3e and f are not, but Figs. 3d and h certainly are. Maybe the authors did not quite see the problem, but rogue waves are supposed to be rare. In the current representation, non-rogue waves actually seem to be the rare ones, even the initial condition already contains tons of rogue waves. The PDF of the linear case should be a parabola in the log representation of Figs. 3(a) and 4(a) according to theory. This case is known as linear interference of long-crested waves in ocean science. Then it would also be helpful to mark the rogue threshold at $4\langle E \rangle$ or 16 in these plots. It's all beautifully there, you just have to plot the correct statistics.

///

In the point 4 above, we have explained in detail the strict equivalence between gaussian statistics for the surface elevation (or electric field), Rayleigh distribution for the wave height (or modulus of the complex electric field), and the exponential distribution for the power (modulus square of the complex field).

Therefore, it does not provide any new physical insight to plot the PDF of $|\psi|$ instead of the PDF of $|\psi|^2$! Moreover, the natural variable that is observed in optics is the Power and it would be artificial and confusing to plot the square root of P instead of P itself.

Reviewer 1 mentions the historical criterion used to define RW (2 or 2.2 times the significant wave height that is 4 times the standard deviation of the SURFACE ELEVATION). In optics this criteria is sometimes used in a confusing way because it is directly applied on the power (by computing the standard deviation of the power !). In the case of Optical power with distribution close to the exponential, the historical threshold used by oceanographers approximately corresponds to 9 times the average of the power in optics (see the Supplementary Material).

Let us emphasize that our goal is not to define RW by using some threshold definition: we examine a more fundamental and general question about the deviations from the central limit theorem that occur from nonlinear propagation in integrable wave systems. As a consequence, the relevant point is the comparison between the initial PDF and the output PDF. The data recorded with our time microscope allows a quantitative comparison between the statistics of nonlinear random waves and the statistics that would arise from the linear superimposition of waves.

As the Reviewer mentions, because we have recorded a large set of data, we do have fluctuations of extreme amplitude in the initial conditions. There is no contradiction between this and the fact that the statistics is Gaussian at initial stage: there is indeed some non-negligible probability that power fluctuations reach 15 times the average value, as shown in Fig. 3.

Finally, we agree with Reviewer 1 : some of the structures displayed on Fig. 3 and 4 do not correspond to the definition of RW (2.2 times H_s with H_s close to 4 times the standard deviation of the surface elevation i.e. the electric field in optics). But these structures are shown here to illustrate the scenario of emergence of Peregrine solitons from initial large fluctuations. We have removed the word RW from the captions and from the comments in the article because it was confusing there.

We understand that some of the readers may be interested in the statistics of the amplitude. We therefore provide a detailed Supplementary Material with the PDF of the amplitude and we also provide the historical threshold of RWs.

//////////

9. The authors say that one-dimensional deep-water waves are ruled by the NLSE. I am not sure about the "deep-water" aspect, but yes, 1D solitons have first been discovered in a canal near Edinburgh, maybe a very deep canal, but what does this have to do with the ocean, the surface of which is undoubtedly two-dimensional? And 2D solitons are unstable. There is another modulation instability. In summary, I think that these are beautiful experiments that should definitely be published. However, the authors should not make the mistake of automatically equating solitons and rogue waves. I think that in particular the introductory paragraph needs to be completely rewritten and that the frequency of "rogue wave" should be brought down to an acceptable 5 or 6 in the manuscript. Then I believe that the authors should refrain from making any statements on ocean physics. This is clearly not their area of expertise, and an expert can immediately sense this in each of their attempted sentences on the subject. Finally, Figures 3 and 4 are clearly missing their marks. It's all about field amplitudes, not intensities, or everything will be rogue.

Our knowledge about hydrodynamics is of course limited but we know that:

- The focusing NLSE describes uni directional propagation of water waves under the assumption of deep water regime (i.e. the wavelength of the carrier wave is much smaller than the water depth) (see any book of hydrodynamics such as Osborne book ref. 19 in the manuscript).
- The first soliton observed in a canal is NOT a deep water wave soliton described by the NLSE. On the contrary, it is a shallow water soliton described by the KDV equation, which is another integrable equation of fundamental importance in Physics.
- The question of directionality of wave trains in ocean has a wide and fundamental importance. Numerous works show that long crested wave trains play a crucial role in oceans. Of course, the ocean is a 2D system and the usual 2D turbulence provide the general theoretical framework to describe this complex system. However, many authors from hydrodynamics consider that 1D water and/or long crested wave trains play a crucial role in ocean (see for example J. Fluid Mech. (2007), vol. 582, pp. 463–472, Gramstad and Trulsen). As non specialist, we do not want to enter in this debate in our manuscript.

As a conclusion about this question, we agree with Reviewer #1 that a strict analogy between ocean and optical fibers cannot be drawn. Our experiments is analogous in its principle to water waves experiments performed in 1D water tank (and not performed in 2D water wave systems). In order to clarify this point, we only mention ocean once in the introduction (RW is a concept that came from oceanographers). We then explain rigorously the one to one correspondence between 1D deep water tank and single mode optical fiber (in the focusing regime). We explicitly say in the conclusion of the paper : “The conclusions which are drawn here in optical fibre experiments well described by the 1D-NLSE cannot be directly extrapolated to phenomena found in the oceans ”

////////

Provided that all these mandatory changes are being implemented, I would lean towards recommending acceptance afterwards, but the introduction currently appears absolutely unacceptable. I think that I offer the authors a golden bridge here how they can turn the introduction around to make this paper worth publication in Nature Communications.

//////////

We thank Reviewer #1 : we think that his comments gave us the opportunity to deeply improve the quality of the introduction and of the conclusion. The context of integrable turbulence being of a general and fundamental importance, we find that it is a good point that it is now introduced from the beginning of the paper.

////////

Reviewer #2 (Remarks to the Author):

Reviewer #2 produced a very positive review, in particular concerning the impact of the paper in the community. We thank Reviewer #2 who emphasizes the fundamental

importance and the novelty of the experimental results (for the first time, the shape of structures emerging from nonlinear random waves can be indeed compared to “breather” solutions of 1D-NLSE).

We have hence revised version of the paper in the following way:

- we have performed modifications of the paper required by Reviewers #1 and #3. We have deeply changed the introduction. In particular we have explained in detail the context of integrable turbulence and we have emphasized the differences between our results and the previous experimental studies in optics.
- However, while performing the modifications, we have payed a particular attention to keep unchanged the main points emphasized by Reviewer #2.

////////////////////////////////////

Answer to Reviewer #3 comments.

Reviewer #3 : “This manuscript shows a measurement of very short pulses of light after propagation through an optical fibre. The technique used to measure picosecond pulses in real time is called "time microscopy" which is already known since 2008 (R. Salem et al. Opt. Lett. 33, 1047 (2008)). This measurement puts into evidence the existence of high intensity pulses which are much higher than the average whenever the average input power exceeds a given threshold. It is already known that the propagation in optical fibres is able to generate the so called optical rogue waves. References are already given in this manuscript. However this work seems to be the first time a direct measurement have been done in this type of system but it is not the first time that optical rogue waves have been directly detected in an optical system (see for example A. Hnilo et al. Opt. Lett, Nov 2011 in mode locked system, Bonatto et al., PRL, July 2011 in laser with optical injection, and Ref.3 in this manuscript includes several other references)

Thus considering that extreme events or optical rogue waves have been detected in optical systems, that optical rogue waves have been "observed" in this type of system, and finally that the technique used here is already known, I think that the manuscript does not have enough significant results to justify publication in Nature Communications.”

////////////////////////////////////

The report of Reviewer #3 contrasts with the two others Reviewers concerning the impact and novelty of the paper. This motivates us to resubmit our manuscript to Nature Communications, and to rewrite critical parts in order to make the impact of the manuscript more evident to a broader range of readers (including, we hope, Reviewer #3). In particular, we have rewritten the introduction, in order to state in an unambiguous way (i) what is new and (ii) in which precise context the work is expected to have a large impact.

Before answering point by point all comments (including more technical ones), we would like first to precise the three following points:

What has been observed for the first time

We have observed – in real time – the SHAPE (i.e., power versus time with a sufficient -sub picosecond- resolution) of optical signals (including optical rogue waves) in this optical fibers experiments (displaying integrable turbulence). We do not pretend that we detected for the first time RW in general (as explicated by Reviewer #3, rogue waves

have already be observed in many different systems). This issue may be due to an inappropriate wording in our introduction. In particular the expression “direct observation” appeared ambiguous (other wordings too, which have been corrected). For instance, previous observations of extreme events in supercontinua and lasers may indeed also be qualified of “direct”, depending on the point of view.

It is very important to note that in the refs as the one cited by Reviewer #3 (A. Hnilo et al. Opt. Lett, Nov 2011), the energy of each pulse emitted by a mode-locked laser is reported. However, the temporal SHAPE (and so the dynamics) is NOT reported in this article.

Up to now, the temporal shape of RW, was only reported in dissipative systems with slow dynamics. As an example, in the second ref. cited by Reviewer #3 (Bonatto et al., PRL, July 2011), the output of the slow dynamics of a laser is recorded with a photodiode. However, the dynamics in this system is described by an ordinary differential equation and the Physics of this dissipative system is of profoundly different nature than the Physics of integrable turbulence.

Our manuscript that deals with integrable turbulence (random initial waves launched into a system well described by the integrable partially differential equation 1D-NLSE) provides for the first time data related to a fundamental conjecture nowadays actively debated : breathers solutions of 1D-NLSE can be considered as prototypes of RWs in some experiments (at least in integrable turbulence phenomena)

In our paper, the core point is the observation of shapes (*with femtosecond resolution, and in “single sweep”* in this system, as would be performed in an oscilloscope in “single” acquisition mode, i.e, without using repetitive measurements). Note that even in the pioneering experiments of Solli et al., the shape of the RWs could not be observed (they used an optical filter and a low bandwidth detector to evidence their existence)

Below (and in the new version of the introduction), we detail why these recordings (using indeed an already existing technique) are of major importance both for the full optical fiber community and for the *integrable turbulence* specialists (as well as – we believe -- for general readers).

Context of the work

From our reading of Reviewer #3 comments, we feel that our manuscript may be misleading for researchers who are at the same time specialists in extreme events (and/or nonlinear dynamics), and not familiar in the new field of research called *integrable turbulence*. This quite recent research field has a deep impact on a large community and is deeply interdisciplinary (optics, hydrodynamics, mathematical physics, integrable wave systems). This also involves concepts belonging to all these fields (in particular non dissipative dynamics). Hence after reading the reports from Reviewer #3, we found important to precise in a detailed way the context of the work (integrable turbulence) in the introduction (which has been thoroughly rewritten).

We think that these clarifications will help to avoid any further ambiguity and allow readers to distinguish the type of dynamics studied here (integrable turbulence) from the dynamics of other very different (and equally important) types of systems as:

- systems with dissipative dynamics (e.g., in lasers)
- systems constantly subjected to noise (see detailed answer below)
- conventional turbulence. Indeed, *Integrable turbulence* is NOT *turbulence*, and the confusion is thus indeed likely to be made also by other readers if some specific precautions are not made (corresponding precisions have been made in the introduction).

Other clarifications are performed in the detailed response below and in the revised manuscript, as the precise *common points and differences* with the water waves context, etc.

Estimated impact of the work

In the general comment of Reviewer #3, it appears that the impact of our work was not evident. Thus we took this remark seriously into consideration while writing the new introduction. Besides, we would like to explicit here what – we think – is of major importance in our results.

A) Obtained data: estimated impact in the large community of Rogue Waves and in particular in the integrable turbulence community.

First, the shapes of the obtained optical signals (including rogue waves) is of central importance, in connection with the debate about the prototypes of RW. As underlined by Reviewer 1, up to now, the idea that “breathers” solutions of 1D-NLSE are prototypes of RW has not been confirmed in experiments. Our work provides the first experimental evidence that the emergence of breather-like solutions of the 1D-NLSE (an equation having a fundamental importance in Physics) is correlated with heavy-tailed deviations from the gaussian statistics in integrable turbulence. As stated by Reviewer 2, we also believe that our “work is of pioneering nature and it opens new possibilities in ultra-short pulse optics” .

B) Experimental strategy: impact for specialists.

The novelty/impact of our experimental strategy (time microscopy) can be viewed from two points of views.

- First, we agree with the fact that time-microscopy is not a new technique. As stated in the paper, this dates back to 1999 (Bennet et al. Ref. 9 of the original manuscript). However we would like to emphasize that the point of the paper is not to invent a new observation tool.
- The important point was more *to find the right strategy*, for obtaining these long-awaited data (signal shapes with sub-picosecond resolution, in single-shot, and high dynamical range).

Moreover, we would like to stress that choosing and developing the “right time-lens-based system” for these measurements is not as trivial that it may have appeared. In particular it involves subtle choices that are not obvious in the optical rogue wave community. For instance, the choice of a time-microscope (versus, e.g, a classical time-lens magnifier), and the choice of the output wavelength (in the visible instead of IR) are crucial for providing extremely high dynamical range (more than 40dB @ 250 fs resolution). This is of major importance (as stated in the paper) for being able to catch in

the same shot the high power peaks, and the moderate power background, as well as, e.g., candidates of peregrine solitons.

Even the potential success of this choice (time microscopy) was – to our knowledge – not obvious at all while reading the literature on rogue waves. We think this may be explained by the level of difficulty for entering such a “time-microscopy” project, with all necessary time-consuming trials, choices, etc.

In this respect, our paper provides – in addition to the fundamental results – crucial guidelines enabling other opticians to reproduce these ultrafast measurements in minimum time (by taking advantage of all subtle experimental choices which we completely detail in Figure 2 and the Method section).

We are quite sure that this milestone will have a deep impact on experimentalists, by motivating reproduction of similar time-microscopy experiments.

C) Impacts on general readers

Though the general reader is not expected to have a background in the debate about prototypes of RW nor in integrable turbulence, we think that our results may have an impact on him, because the paper shows for the first time “snapshots” of sub-picosecond optical rogue waves, in the sense that: (i) recordings are made in single shot (i.e., without needing repetitive measurements), and (ii) with a sub-ps temporal resolution which gives access to the temporal rogue wave shapes (as opposite to the – nevertheless also very nice – experiments quoted by Reviewer #3).

Last but not least, to our knowledge it is the first ultrafast dynamical system (for which single-shot recording is necessary) that has been ever been analyzed using a time-lens system (Up to now time lens was demonstrated as a technique but not used to solve a fundamental question). We think that – though it is not the main objective of our paper – this may contribute to future widespread of time lens and time-microscope applications.

As a conclusion, beyond the study of RWs in integrable turbulence, our work has a deep impact on the whole community of Optics as it is the first single-shot observation of **random** fluctuations having the ultrafast (sub-picosecond) time scale commonly observed in optics (mode locked lasers, solitons in optical fibers...).

DETAILED ANSWER TO REVIEWER #3 COMMENTS

Note on the rogue wave examples quoted by Reviewer #3

“[...] it is not the first time that optical rogue waves have been directly detected in an optical system (see for example A. Hnilo et al. Opt. Lett, Nov 2011 in mode locked system, Bonatto et al., PRL, July 2011 in laser with optical injection, and Ref.3 in this manuscript includes several other references)”

The word “RW” has been used in an extremely wide amount of optical papers. As a consequence, the phrase RW nowadays refers to different situations that must not be confused and mixed:

- A one to one correspondence between optics and hydrodynamics is provided by the fundamental 1D NLS equation. Experiments performed in optical fibers thus have an historical importance in the study of rogue waves (see the historical paper of Solli et al.

and all the papers of Dudley et al about RW such as refs. 20, 27 of the manuscript). While they are described by 1DNLSE, the experiments performed in optical fibers are extraordinary laboratories to investigate the physics of the known structures such as fundamental solitons, Akhmediev breathers, Peregrine solitons (see ref. [29-32] for example). In optical fibers, the natural time scale involved in these structures is the picosecond (or less) time scale.

-These experiments performed in nearly Hamiltonian and integrable systems cannot be generally compared to dissipative systems such as lasers. In such a case, the Physics is of profoundly different nature : for example solitons on finite background are not solutions anymore. In some of the dissipative systems, the dynamics can be very slow and we agree that it is then very easy to observed the dynamics with a standard photodetectors.

The exact solutions of 1DNLSE (Akhmediev breathers, Peregrine solitons...) play a fundamental role because they are now considered as prototypes of RW. This conjecture is shared by a large community and up to now, there is no experimental demonstration that these structures can indeed emerge from from the nonlinear propagation of random waves.

As a matter of fact, it is not an acceptable argument to say that previous observations of slow extreme events in laser make the observation of fast coherent structures in fiber optics experiments described by 1DNLS less fundamental, less important or even “not new”.

On the contrary, our observation provide to the community the first time-resolved data (i.e., power evolutions with sub-picosecond resolution) allowing to examine the question of the emergence of the solutions of NLSE from the propagation of RANDOM WAVES. As emphasized by Reviewers 1 and 2, our study is of fundamental importance and is at the front edge of experimental science. We explain in our manuscript that the time microscope has been demonstrated as a possible technique in 1999 (and not in 2008 as mentioned by Reviewer 3). However, it is important to note that up today, this technique has never been used in fundamental study devoted to nonlinear dynamics and nonlinear random waves. Note again that in the pioneering work of Solli et al, the fast temporal evolution (at sub-picosecond scale) was not recorded : the signal was optically filtered before being recorded with a standard (nanosecond) photodetector, and an oscilloscope.

The proper context of our work is the so-called integrable turbulence introduced by Zakharov. We understand from the questions of the Reviewer 3 (and from the remarks about our introduction of the Reviewer 1), that the importance of our work in this specific framework was not clearly stated in the initial manuscript. We have therefore completely changed the introduction of the paper in order to explain clearly why the results presented here are of fundamental importance. Moreover, our work is analogous in its principle to 1D water tank experiments but we want to emphasize here that our goal is not to develop an extensive analogy between ocean waves and optical fiber optics. We also clarify this point in the revised manuscript.

/////

Question 1 of Reviewer 3 : “Some general statements in the introduction relating rogue waves observed in the ocean with extreme events in optical systems are in

my opinion too much speculative. I think that equivalence between two very complex dynamical systems can be established only through measurements that are not available in the comparison between the ocean and optical systems. Templates, bifurcation diagrams and other techniques can not be available. Therefore I prefer that the reader will not imagine that optical rogue waves are the same as rogue waves in the ocean.”

///////

We do agree with the Reviewer 3 that great care has to be taken in the analogy between optics and hydrodynamics. However there is a perfectly rigorous one-to-one correspondence between optics and hydrodynamics : at leading order, the physics of 1D wave trains in the so-called deep water approximation is described by the focusing 1D NLSE (see ref. 21 of the manuscript for example). As a consequence, our experiments are analogous in their principle to the experiments performed with random water waves in 1D water tank (Onorato et al, PRE, 2004, ref 54). However, note that our experiments are much more nonlinear than hydrodynamical experiments.

We explain these points in detail in the new version of the manuscript. In particular in the conclusion, we added the paragraph :

“The conclusions which are drawn here in optical fibre experiments well described by the 1D-NLSE cannot be directly extrapolated to phenomena found in the oceans. Let us emphasize however, that the principle of our experiment is identical to the one of some previous experiments performed in one-dimensional deep water tank [54]. Starting from random initial conditions, those hydrodynamical experiments have also demonstrated the formation of heavy-tailed statistics. Note finally that the extreme amplitudes observed in our optical fibres experiments could not be observed in water waves that are strongly limited by the wave breaking phenomenon [1, 55, 56]. “

//////////

Question 2 of Reviewer 3 : “ The manuscript says that "common ocean waves are weakly nonlinear random objects having nearly Gaussian statistics". I suppose that researchers in oceanography will never accept such statement. In fact most of them tried to developed models that will not give Gaussian statistics because Gaussian models will give perfect symmetric waves which clearly are not observed very often in a "moving" sea!! “

///////

We have removed this first sentence of our manuscript because we understand from the comment of Reviewer 3 (and Reviewer 1) that it was confusing.

However, researchers in oceanography do agree with this statement if it is fully explained : ON AVERAGE, the ocean is weakly nonlinear. The strength of the nonlinearity of deep water waves is measured by the steepness (product of the amplitude by the wave vector). On average, the steepness in deep water waves is around 0.1 because of the wave breaking. The wave breaking thus play the role of a cut-off mechanism that prevents the occurrence of large nonlinearities (on average) (see refs 55, 56). Of course, this does not mean that nonlinearities do not play any role (in particular in the formation of RW).

We understand that this important point was unclear in our initial manuscript. We have clarified this and we have also added a comment about the wave breaking in ocean (our experiments is in a strongly nonlinear regime that is not achievable in water waves)

//////////

3) Reviewer 3 : “An oceanic rogue wave is not necessarily a spatially localised peak. The definition of a rogue waves for people working on oceanography is based only on the high of the wave compared to the average. There is no mention about the propagation distance. I do not think in optics it requires such localisation. On the other hand the measurement presented here is only in time and not in space because the system is by definition a 1D system. The spatial coordinate being the propagation one is directly related to time, then I do not understand. What is the meaning of localisation? A short pulse in time is essentially the same as localisation in the spatial coordinate.”

//////////

We fully agree with the referee : the historical definition of RW from a threshold does not imply the localization of the structure. However, a large part of the recent literature considers that one of the possible scenarii of RWs formation is the emergence of localized (in space and in time) solutions of NLSE (Peregrine soliton is one of the example).

However, in our manuscript, the word “localized” means “localized in time”. Indeed, from the conceptual point of view, the variable of evolution is z and the equivalent of the dynamical “space” is the time (the second derivative in NLSE). Most of the theoreticians work with the 1D-NLSE under the following form: $i \partial \psi / \partial t = \partial^2 \psi / \partial x^2 + |\psi|^2 \psi$. A short pulse in our experiments therefore corresponds to a pulse that is localized in space. We have removed the ambiguous use of the word localization in the new version of the manuscript.

Moreover, we have added a new detailed section in the Supplementary Material in order to present the values of the standard threshold of RW found in our experiments.

//////////

4) Reviewer 3 : “The amplification of a pulse giving rise to high intensity is not surprising in a nonlinear system like this one.”

//////////

This claim from the Referee 3 is surprising and unclear. There is no real “amplification” in our system. Of course, as our wave system is described by the focusing 1D-NLSE, one expects focusing phenomena. However, we investigate here new scenarii that are not well known in the community : in particular, the universal emergence of the Peregrine solitons from the propagation of single hump has been recently demonstrated by the mathematicians Bertola and Tovbis (see the details in our manuscript and ref. 53 of the manuscript). We demonstrate for the first time that solutions of 1DNLSE such as solitons on finite background can emerge from the propagation of random waves. This is a fundamental and completely new result. Maybe the referee is not surprised by the result but to the best of our knowledge, one can find many articles with this conjecture without any experimental observations.

As pointed out by the Reviewers 1 and 2, we believe that our experimental results are of crucial importance at the front edge of the nonlinear dynamics and integrable turbulence.

//////////

5) Reviewer 3 : "If I understood correctly random pulses in time at very well defined optical frequency are propagated through the optical fibre. The output shows extreme events. However the manuscript is saying that the system is turbulent. There is no proof at all along the manuscript that the system is turbulent. What definition of turbulence have been used? What measurements indicate or put in evidence the turbulent character of the dynamical behavior of the system? I did not see any measurement of the loss of spacetime correlation or an inverse cascade in the spectrum? It seems to me that such evidence is not presented here nor in the experimental results or even in the numerical ones. "

////

First, our initial manuscript was probably unclear : we do not launch "random pulses at very well define optical frequency" in the fiber. As explicitly said in the manuscript, we use partially coherent waves continuously emitted by ASE source (of course the power exhibit large fluctuations but these fluctuations are not "pulses" in the usual sense used in optical experiments, they rather represent slow random modulations of the light power). The time scale of the power fluctuation is controlled by the spectral width of the spontaneous emission source.

We thank the referee 3 for this comment because we have understood that we have to be more pedagogical about the situation of the context of our work. The correct framework of our study is INTEGRABLE TURBULENCE. This word proposed by Zakharov includes all complex dynamics that can be observed in integrable systems (or closely integrable systems) with random initial conditions. As a matter of fact, our optical fiber system is not turbulent in the usual meaning : in particular, there is no energy cascade neither inverse cascade in the spectrum because there is no resonant interactions.

However, an important point is the well known spectral broadening phenomenon (i.e. decreasing of the coherence time with power). This point is extremely well documented in previous works and we have put experimental and numerical spectra in the initial version of the supplementary material (section II). We do not think that this point deserves a Figure in the main article as no new idea is provided here.

In our initial manuscript, we thought that this point was clear because many references were given about integrable turbulence. We have decided to clarify the context of our work and the exact meaning of "integrable turbulence" in the completely new and longer version of the introduction.

////////

6) Reviewer 3 : "In page 5 the manuscript reads: "The emergence of coherent structures is a general mysterious feature of stochastically driven processes....." I did not get such statement. The generation of coherent structures from noise is a general behaviour in many physical systems, being lasers one of the most well known examples in optics where a coherent beam grows from spontaneous emission. "

////

We agree with Reviewer 3 on this point, and removed this text. Originally, we intended to stress that self-organization from a noisy initial condition in **non-dissipative** systems was a largely open problem, which is true (see, e.g., Solli et al., Nature Photonics, 2012). We agree that the statement is not relevant in all contexts. In particular, in dissipative systems (as lasers, pattern-forming systems), self-organization from noise as well as noise-sustained structures etc. have been largely studied. Even textbooks exist on these topics, and the question is largely less open. The best was to remove this piece of text (which was not necessary and ambiguous).

Moreover, this comment and the following ones of the reviewer 3 (see below) pointed out an ambiguity in our description of the role of noise in our system. At this point we would like to re-precise the role played by noise in this system (in the experiment and in the model):

- We use a **random initial condition** (i.e., at the input of the fiber).
- No noise is added nor “affects” the electric field during the propagation of the fiber in our experiments. The propagation itself is purely deterministic.

Hence our phrase “stochastically driven” was not appropriate. We have removed the related ambiguous sentences and payed a particular attention to clarify this point in the new manuscript.

//////////

7) Reviewer 3 : “Finally I do not understand the comparison with the NLSE. If the NLSE describes the system and the system is driven by a random source, then it is not enough to put a random initial condition but it requires a noise term included into the equation. There is a fundamental difference in dynamics between the meaning of a noise driven system with respect to a random initial condition even if the system is conservative. “

////

There is indeed a fundamental difference between noise driven systems (with noise that adds to the optical field during the propagation in the fiber for example) and deterministic systems driven by random initial conditions. We thought our initial manuscript was clear with the following sentences :

- “ Randomness of the initial condition is known to play a crucial role in the generation of Rws”
- “ Starting from random fluctuations having typical time scale around 5-10 ps [Fig 1(a)], those extreme peaks are also found to be extremely narrow, with time scales of the order of several hundreds of femtoseconds”
- “ More precisely, the random waves used as initial conditions in our experiments are partially coherent light waves emitted by a high power Amplified Spontaneous Emission”
- “ Starting from random light propagating with a mean power of 4 W in the fibre,

huge RWs having peak power that exceeds 300 W”

-“ Fig. 4 shows a picture of typical random fluctuations of the optical power that are found at the input and output ends of the optical fibre.”

- “ On the contrary, the initial conditions in our experiments are designed to be “ocean-like” random waves”

We thank Reviewer 3 who gives us the opportunity to avoid a misleading ambiguity : we remove the ambiguous word “stochastically driven” or “noise-driven”. Our experiments is based on the propagation of partially coherent waves (random waves with finite width of the optical spectrum).

Reviewer #1 (Remarks to the Author):

Having read the reply to my remarks, let me state very clearly that I believe that the authors have done an extraordinary job in turning this paper around and making full use of the constructive part of the criticism raised in the first round of review. The authors contradict a few aspects of my review, but I have to confess that they provide very good reasoning to do so. I certainly do not insist on being right in all my criticism. Nevertheless, there are a few minor aspects that require attention by the authors.

1. The wording "breathers-like" is a bit confusing. I think it should be singular, i.e., "breather-like".
 2. Abstract: "Ultrafast structures with extremely high power are generated..." I think that this sentence will be misunderstood. The resulting powers are actually not that impressive, given that a simple mode-locked oscillator already produces a megawatt peak power and that the ELI initiative now heads towards tens or even hundreds of petawatts whereas the authors probably have a kilowatt at best. I think this sentence should be changed to indicate more clearly that it is actually the contrast between peak powers of individual events and the background level that is extreme here.
 3. Sentence "RWs have been first studied in Oceanography where they have been identified as being rare events that may significantly damage ships in some circumstances."
I am not quite sure how this sentence would be received by a ship captain who encountered a rogue wave. Rogue waves can actually sink ships, and the circumstance is poorly understood on when the encounter may be deadly, other than, maybe, that it was in a severe storm. And the ocean is the worst system to actually study rogue waves as they are so rare.
 4. Concerning the integrable turbulence, let me first fully agree with restricting the analysis to this type of turbulence. Nevertheless, I do not agree with the notion that this is a relatively new direction of research. Zakharov has already published on this in the 1980s, see, e.g., V. E. Zakharov et al. Soliton turbulence. JETP Lett., 48, 79-82 (1988). He did not use the exact term "integrable turbulence", but it is a fairly old idea. Akhmediev only followed up on these ideas and yet calls it another name.
 5. In the fourth paragraph, partially coherent rogue waves are mentioned. In my interpretation of the authors' categorization, there are coherent rogue waves (in optical fibers) and incoherent ones (in the ocean). It's not quite clear to me what the hybrid refers to. The term also appears later on. It needs to be defined precisely what is a partially coherent light wave. In my understanding, the light waves discussed in this manuscript are actually fully coherent, meaning that they are reproducible from shot to shot if only input conditions are exactly the same.
- From my point of view, the paper is good to be published. The necessary edits are absolutely minor. As I think that these findings really constitute a highly important missing link between the so-called coherent rogue waves and the real incoherent ones, I strongly support publication now, with the only prerequisite of abandoning the concept of partially coherent rogue waves once and forever.

Reviewer #2 (Remarks to the Author):

It is clear from the discussion between the referees and the authors of this manuscript that the rogue wave research in optics is presently at its highest point. Therefore introduction of new tools for their investigation is vitally important. I can see the use of the "time microscope" for these studies as the main achievement of the present work.

This tool allows firstly to solve some of the existing problems related to optical rogue waves and necessarily will pose new questions.

The whole lengthy discussions are exactly about clarifying these problems that the science of rogue waves brings us today. As such, the paper is a new fundamental step that

creates new interest in this quickly raising branch of optics.

The possibility of single shot studies in optics at picosecond durations in real time is a major step forward that elevates optical measurements to a new qualitative level.

Looking at this manuscript from this perspective, I can only say that it must be published.

Most of the issues raised by the referees are caused by the natural desire of scientists to solve all problems in one go. Clearly, this is not possible. Especially when a crucial step forward is done. I can see this work as the keystone that will provoke new studies with the use of similar or better equipment in future.

Being the first is apparently a difficult step. No doubts that the attention of the whole community working on the subject of optical rogue waves will be concentrated on this from the day when the work will appear in print. Taking this point into account, the work must be published in Nature Communications.

Most of the concerns of the referees are pretty much wisely addressed by the authors. To my view, the paper cannot be improved more. The views vary and there will be always objections when the principally new step is done.

There are quite a few deviations in discussing water waves but this is not what has been done in the work. The theoretical aspects are also still an open and active field of research. Some problems may require decades to be solved. Experimental breakthroughs are crucial for doing this.

The authors are suggesting a revolution on the experimental side of the most active branch of optics and this aspect of their work should be appreciated to the full extent.

Answer to Reviewer 1

We acknowledge the extremely positive comments of the Reviewers 1 and 2. We also thank the Reviewers who gave comments and questions that have significantly contributed to the improvement of the quality of the manuscript.

Reviewer recommendation 1. The wording "breathers-like" is a bit confusing. I think it should be singular, i.e., "breather-like".

We agree and have used only singular form, i.e. "breather-like".

Reviewer recommendation 2. Abstract: "Ultrafast structures with extremely high power are generated..." I think that this sentence will be misunderstood. [...] I think this sentence should be changed to indicate more clearly that it is actually the contrast between peak powers of individual events and the background level that is extreme here.

We have replaced the sentence that now reads :

Ultrafast structures having peak powers much larger than the average optical power are generated from the propagation of partially coherent waves ...

Reviewer recommendation 3. Sentence "RWs have been first studied in Oceanography where they have been identified as being rare events that may significantly damage ships in some circumstances."

I am not quite sure how this sentence would be received by a ship captain who encountered a rogue wave.

We have removed this sentence. We have just added "First studied in Oceanography" at the beginning of the first sentence of the introduction.

Reviewer recommendation 4. Concerning the integrable turbulence, let me first fully agree with restricting the analysis to this type of turbulence. Nevertheless, I do not agree with the notion that this is a relatively new direction of research. Zakharov has already published on this in the 1980s, see, e.g., V. E. Zakharov et al. Soliton turbulence. JETP Lett., 48, 79-82 (1988). He did not use the exact term "integrable turbulence", but it is a fairly old idea. Akhmediev only followed up on these ideas and yet calls it another name.

Although "integrable turbulence" and "soliton turbulence" have close designations, they refer to two fields having completely different natures. As stated in our manuscript, the field of "integrable turbulence" deals with nonlinear random wave systems that are ruled by integrable equations. On the other hand, the field of "soliton turbulence" deals with nonlinear evolution of random fields in wave systems that are ruled by equations that are NOT integrable (see the abstract of Zakharov, JETP Lett., 48, 79-82 (1988)). The physical features that are found in integrable turbulence and in soliton turbulence are of profoundly different natures. Nevertheless it is not necessary to insist on the relative novelty of the field of integrable turbulence. We also fully agree that Zakharov has introduced the idea (and the word integrable turbulence). We acknowledge it explicitly and we cite the main articles in which this word has been used.

We have removed “new field of research” and our sentence now reads :

“The problem of random (or partially coherent) inputs in wave systems described by integrable equations such as 1D-NLSE enters within the fundamental framework of the so-called “integrable turbulence” introduced by V. E. Zakharov [34-39]

Reviewer recommendation 5. In the fourth paragraph, partially coherent rogue waves are mentioned. In my interpretation of the authors' categorization, there are coherent rogue waves (in optical fibers) and incoherent ones (in the ocean). It needs to be defined precisely what is a partially coherent light wave.

We understand from this comment that the notion of Partially coherent waves has to be more clearly defined in the manuscript. This concept is very common in the field of statistical optics and it does not imply that the propagation is not deterministic. Partially coherent waves represent a category of random waves having a spectrum of finite width (partially coherent waves has to be distinguished from white noise or from supercontinuum).

In order to clarify this point, we add the following sentences in the introduction :

Partially coherent waves correspond here to random waves whose optical spectral width is finite and small in comparison with the carrier wave frequency [43]. Assuming independent random Fourier components the statistics of the partially coherent field is Gaussian [43].

and

We study experimentally the changes of the dynamics and of the statistics arising from the deterministic and nonlinear propagation of partially coherent waves in optical fibre.

In the Section Methods - Numerical Simulations details, we add the sentence :

“Note finally that we consider here the deterministic 1D-NLSE (whithout any additionnal noise occurring along the propagation inside the fibre)”